# Brain Morphometry and Cognitive Features in the Prediction of Irritable Bowel Syndrome

**DOI:** 10.3390/diagnostics15040470

**Published:** 2025-02-14

**Authors:** Arvid Lundervold, Ben René Bjørsvik , Julie Billing , Birgitte Berentsen , Gülen Arslan Lied , Elisabeth K. Steinsvik , Trygve Hausken , Daniela M. Pfabigan , Astri J. Lundervold 

**Affiliations:** 1Department of Biomedicine, University of Bergen, 5020 Bergen, Norway; arvid.lundervold@uib.no; 2Medical-AI, Mohn Medical Imaging and Visualization Centre, Department of Radiology, Haukeland University Hospital, 5021 Bergen, Norway; ben.bjorsvik@student.uib.no; 3Department of Biological and Medical Psychology, University of Bergen, 5020 Bergen, Norway; julie.billing@uib.no (J.B.); daniela.pfabigan@uib.no (D.M.P.); 4Department of Clinical Medicine, University of Bergen, 5021 Bergen, Norway; birgitte.berentsen@uib.no (B.B.); trygve.hausken@uib.no (T.H.); 5National Center for Functional Gastrointestinal Disorders, Department of Medicine, Haukeland University Hospital, 5021 Bergen, Norway; gulen.arslan@uib.no (G.A.L.); elisabeth.kjelsvik.steinsvik@helse-bergen.no (E.K.S.); 6Center for Nutrition, Department of Clinical Medicine, University of Bergen, 5021 Bergen, Norway

**Keywords:** irritable bowel syndrome, structural MRI, brain morphometry, cognition, supervised classification, machine learning, open-source

## Abstract

**Background/Objectives:** Irritable bowel syndrome (IBS) is a gut–brain disorder characterized by abdominal pain, altered bowel habits, and psychological distress. While brain–gut interactions are recognized in IBS pathophysiology, the relationship between brain morphometry, cognitive function, and clinical features remains poorly understood. The study aims to conduct the following: (i) to replicate previous univariate morphometric findings in IBS patients and conduct software comparisons; (ii) to investigate whether multivariate analysis of brain morphometric measures and cognitive performance can distinguish IBS patients from healthy controls (HCs), and evaluate the importance of structural and cognitive features in this discrimination. **Methods:** We studied 49 IBS patients and 29 HCs using structural brain magnetic resonance images (MRIs) and the Repeatable Battery for the Assessment of Neuropsychological Status (RBANS). Brain morphometry was analyzed using FreeSurfer v6.0.1 and v7.4.1, with IBS severity assessed via the IBS-Severity Scoring System. We employed univariate, multivariate, and machine learning approaches with cross-validation. **Results:** The FreeSurfer version comparison revealed substantial variations in morphometric measurements, while morphometric measures alone showed limited discrimination between groups; combining morphometric and cognitive measures achieved 93% sensitivity in identifying IBS patients (22% specificity). The feature importance analysis highlighted the role of subcortical structures (the hippocampus, caudate, and putamen) and cognitive domains (recall and verbal skills) in group discrimination. **Conclusions:** Our comprehensive open-source framework suggests that combining brain morphometry and cognitive measures improves IBS-HC discrimination compared to morphometric measures alone. The importance of subcortical structures and specific cognitive domains supports complex brain–gut interaction in IBS, emphasizing the need for multimodal approaches and rigorous methodological considerations.

## 1. Introduction

Irritable bowel syndrome (IBS) represents a prevalent and complex functional gastrointestinal (GI) disorder, affecting approximately 10% of the global population [1]. The syndrome is clinically defined by a characteristic symptom pattern, namely recurrent abdominal pain associated with defecation, accompanied by alterations in bowel habits [2], and can be divided into clinical phenotypes based on predominant bowel patterns [3] and overall symptom severity [4]. The clinical presentation is heterogeneous, with experiences ranging from mild discomfort to severe symptoms that substantially impair quality of life and daily functioning [4]. Notably, women are disproportionately affected, a difference that appears to arise from a complex interplay of biological factors (including hormonal influences), healthcare-seeking behaviors, and sociocultural determinants [5,6,7,8]. Such epidemiological patterns highlight the multifactorial nature of IBS and underscore the importance of considering both biological and psychosocial factors in its study and treatment.

A bidirectional relationship between GI symptoms of IBS and psychological functioning is well-documented [9]; while GI symptoms can trigger or exacerbate psychological distress, anxiety and depression may in turn amplify the intensity and frequency of abdominal pain [10]. Recent research has expanded this psychobiological framework to include cognitive function, revealing a more nuanced picture of brain–gut interactions in IBS. Although cognitive impairments have been demonstrated at the group level [11,12], these deficits seem to characterize specific subgroups rather than being a universal feature of IBS [9,13]. This heterogeneity in psychological and cognitive presentations aligns with contemporary models of the gut–brain axis [14,15], which conceptualize IBS as a disorder of disrupted neural–enteric communication. In these models, the brain serves as the central integration hub for processing and interpreting the complex array of visceral signals, emotional responses, and cognitive processes that may be involved in IBS.

The relationship between brain structure and cognitive function has evolved from simple localization models to more sophisticated network-based frameworks [16,17]. This network perspective gained particular relevance for understanding IBS through Mayer et al.’s [18] seminal paper in 2015, which proposed that alterations in brain networks could directly influence multiple cognitive domains in IBS patients (see also [19]). Recent empirical support for this systems-level approach comes from Li et al. [20], who identified several associations between symptom severity and regional brain volumes, including positive correlations with subcortical structures (globus pallidus, caudate, and putamen) and negative correlations with cortical regions (anterior cingulate, dorsolateral prefrontal cortex, and anterior and mid-cingulate cortices) and subcortical areas (anterior insula, hippocampus, parahippocampal cortex, and thalamus). One of their findings is of special interest to the present study; they also showed that these brain regions were linked to cognitive performance on tests of language skills and memory function.

Studies of abdominal pain and visceral stimulation have consistently demonstrated the involvement of distributed brain networks, encompassing both cortical and subcortical structures [21,22]. Building on this network perspective, Skrobisz et al. [23] conducted a comprehensive morphometric analysis in patients with non-specific digestive disorders, including IBS. Using FreeSurfer software (version 6.0.1), they analyzed 36 brain regions, including subcortical, cortical, and global measures derived from structural magnetic resonance imaging (MRI). Their univariate analyses revealed a reduced thalamic volume in IBS patients compared to healthy controls, though volumes remained larger than in patients with inflammatory bowel diseases. While these findings suggest structural brain differences in IBS, univariate approaches may not capture the full complexity of brain–gut interactions. Therefore, our study builds upon Skrobisz et al.’s work in two key ways. First, we examine the robustness of their findings by comparing analyses using both FreeSurfer v6.0.1 and a more recent version, allowing us to differentiate between software-dependent and true biological effects. Second, we extend beyond univariate analyses by implementing multivariate approaches, including supervised machine learning techniques, to capture complex patterns in brain morphometry that might better characterize IBS. This dual approach—methodological validation and advanced pattern analysis—aims to provide a more comprehensive understanding of the structural brain differences associated with IBS.

Finally, responding to Skrobisz et al.’s [23] call for integrating clinical measures, we investigated whether combining cognitive performance data with morphometric features would enhance the accuracy of IBS versus HC classification.

Our study has four key aims, as follows:AWe aim to replicate the morphometric differences between IBS patients and HC reported in [23] using the same FreeSurfer software version (FS 6.0.1) and a similar univariate analysis approach as in the original study.BWe aim to evaluate consistency between FreeSurfer versions by comparing morphometric segmentation outcomes from version 6.0.1 (used in [23]) and version 7.4.1 in our dataset (n=78).CWe aim to assess whether morphometric features from FS 7.4.1 (both cross-sectional and longitudinal analyses) can differentiate IBS from HC groups through the following means: (i) univariate group comparisons, (ii) multivariate analyses incorporating feature covariance, (iii) machine learning classification, and (iv) feature importance analysis of successful classifications.DWe aim to determine whether incorporating cognitive performance data enhances the morpho-metric-based machine learning classification, and if so, we aim to identify the most discriminative features between IBS and HC groups.

## 2. Materials and Methods

### 2.1. Participants

This study is part of the Bergen Brain-Gut project, a prospective clinical investigation conducted at Haukeland University Hospital, Norway (2020–2022; protocol detailed in Berentsen et al. [24]). We enrolled 78 participants (49 IBS patients and 29 healthy controls [HCs]), all of whom were ≥18 years old. Recruitment occurred through media advertisements, informational flyers, and direct referrals from the hospital’s outpatient clinic. A trained nurse screened all candidates using standardized inclusion and exclusion criteria (Table 1). Eligible participants underwent comprehensive assessment including gastrointestinal measures, psychometric testing, and multiparametric magnetic resonance imaging (mpMRI).

The determination of our sample size balanced multiple considerations. Although we did not conduct an a priori power analysis due to limited effect size data on brain morphometric differences in IBS at study inception, our sample size met or exceeded those of comparable neuroimaging studies on functional GI disorders [23,25,26,27]. We included only participants with complete key measures and high-quality MRI scans suitable for automated brain segmentation, optimizing data quality while maximizing sample size.

The Bergen Brain-Gut project’s initial cohort consisted of 85 subjects with baseline MRI scans. Our final analytical sample of 78 participants (92% inclusion rate) was determined by predefined criteria. The seven excluded participants consisted of four subjects lacking RBANS test results, one participant was excluded due to non-Norwegian language proficiency affecting cognitive testing validity, and two subjects had incomplete datasets (one IBS patient, one healthy control). These exclusions were based on missing data or predefined quality criteria rather than post hoc selection, and the balanced distribution across patient and control groups suggests minimal risk of systematic bias.

### 2.2. Measures

Age and sex (not genetically verified) were self-reported by the participants at baseline.

#### 2.2.1. The IBS-Severity Scoring System (IBS-SSS)

The IBS-Severity Scoring system is a questionnaire used to assess the severity and frequency of GI-related IBS symptoms [28]. The questionnaire includes five items related to (i) abdominal *pain intensity,* (ii) abdominal *pain frequency*, (iii) abdominal *distention/bloating*, (iv) dissatisfaction with *bowel habits*, and (v) interference with *quality of life*, all over the past 10 days. IBS-SSS scores range from 0 to 500, with higher scores indicating greater symptom severity. The maximum score for each question is 100. A sum of scores <75 is used to define “no or minimal problems”, and the scores in the ranges [75,175), [175,300], and >300 are defined as “mild”, “moderate”, and “severe” IBS symptoms, respectively [28]. In the present study, an IBS-SSS score ≥175 was used as the inclusion criteria for the IBS group. Of the 29 HC participants, 26 (89.7%) obtained an IBS-SSS score below 75 (lowest level), while 0 (0.0%) reported scores between 75 and 175 (mild level). The median IBS-SSS score for the HC group was 21.0 (Interquartile Range IQR: 9.8–39.8), with a maximum score of 69.0 in this group.

#### 2.2.2. Repeatable Battery for the Assessment of Neuropsychological Status (RBANS)

All participants performed the Norwegian version of the Repeatable Battery for the Assessment of Neuropsychological Status (RBANS version A). It was administered by a nurse trained by a clinical neuropsychologist, following the instructions of the test manual [29]. RBANS was included to provide a quick and comprehensive assessment of cognitive function. It takes less than 30 min to complete and has been shown to be sensitive to mild cognitive impairment, with good reliability and validity. The following five key cognitive domains are calculated: (i) *immediate memory*, (ii) *visuospatial/constructional skills*, (iii) *language*, (iv) *attention*, and (v) *delayed memory*. Each of the first four domains comprises two subtests, and the results on the four memory tests are summed and combined with the results on a recognition test to obtain the total raw score for the *delayed memory* index. Test scores, expressed as age-corrected index scores, are included in the present study. The index scores have a mean value of 100 and a standard deviation of 15 and are based on performance in a normative group matched to 2012 population statistics in Norway, Sweden, and Denmark. A full-scale RBANS score gives an overall measure of cognitive function across all the five indexes.

### 2.3. MRI Data Acquisition

All neuroimaging data were acquired using a 3 Tesla Siemens Biograph mMR PET/MRI scanner (Siemens Healthineers, Erlangen, Germany) equipped with a standard 12-channel head coil. The comprehensive multiparametric imaging protocol consisted of five sequences: a three-dimensional (3D) T1-weighted Magnetization Prepared Rapid Gradient Echo (MPRAGE) (TA = 5:35 [min:sec]), T2-weighted structural imaging (TA = 5:12), gradient echo (GRE) field mapping (TA = 0:54), a resting-state functional MRI using echo-planar imaging (EPI) with integrated motion correction (TA = 9:48), and diffusion-weighted imaging with 30 gradient directions and three b-values (TA = 8:34). The total examination time was approximately 45 min.

For the current morphometric analyses, we utilized only the high-resolution T1-weighted images, acquired using a 3D MPRAGE sequence. The acquisition parameters included a spatial resolution of 1.0 mm isotropic (1×1×1 mm^3^) across 192 sagittal slices, with a repetition time (TR) of 2500 ms, an echo time (TE) of 2.26 ms, and an inversion time (TI) of 900 ms. The field of view (FOV) was set to 256 × 256 mm^2^ with a corresponding matrix size of 256 × 256, and parallel imaging was employed using GRAPPA with an acceleration factor of 2.

Figure 1 shows a representative T1-weighted image from our dataset, demonstrating the high tissue contrast necessary for accurate morphometric analysis. The corresponding FreeSurfer-generated segmentation mask, which forms the basis for our morphometric measurements, is illustrated in Figure 2. These images exemplify the quality standards maintained throughout our dataset.

### 2.4. Brain Morphometry Analysis Using FreeSurfer

Image processing and morphometric analyses were performed using FreeSurfer (https://freesurfer.net (accessed on 11 February 2025)), a widely-validated open-source software suite used for analyzing brain MRI data [30]. To address both methodological and biological questions, we conducted parallel analyses using two FreeSurfer versions: version 6.0.1, which was employed in the reference study by Skrobisz et al. [23], and the current version 7.4.1.

The evolution of FreeSurfer’s capabilities is particularly relevant to our investigation of brain structure in IBS. Version 7.0 (July 2020) introduced significant improvements in subcortical segmentation accuracy, while version 7.4.1 (June 2023) further enhanced the precision of limbic system structures, notably the hippocampus and amygdala. Additionally, version 7.4.1 provides superior compatibility with multimodal imaging data and implements refined longitudinal processing algorithms. Since our multimodal MRI examinations were part of a longitudinal IBS intervention study (Berentsen et al. [24]), we also used the longitudinal stream capability of FreeSurfer 7.4.1 to compare baseline longitudinal analysis with a cross-sectional analysis of the first MRI examination.

For both versions, we focused on the automated segmentation of subcortical structures using FreeSurfer’s aseg pipeline, which identifies and quantifies the volume of distinct brain regions (detailed in Table A1). This dual-version approach serves two purposes: first, it enables direct comparison with Skrobisz et al.’s [23] findings, and second, it allows us to assess the impact of software evolution on morphometric measurements in a fixed dataset and differences in cross-sectional and longitudinal stream analysis, in order to discriminate HC and IBS from brain morphometric features. This methodological consideration is crucial, as previous studies have demonstrated that version-dependent variations in automated segmentation can significantly influence morphometric results [31,32,33,34,35,36]. By analyzing our data with both versions, we can distinguish between genuine biological differences and methodologically-induced variations in brain morphometry.

The enhanced accuracy of version 7.4.1 is particularly relevant for our investigation of IBS, as it provides more reliable quantification of brain regions implicated in visceral sensation, pain processing, emotional regulation, and cognitive function.

We will also like to add that in vivo brain segmentation technologies develop very fast. Recently (November 2024), the FreeSurfer 8.0.0-beta version enabled histological super granularity with identification and volume measurements from more than 300 distinct regions per hemisphere (cf. Figure A2). The aseg mask provides less than 40 brain regions and their volumes within the intracranial space.

### 2.5. Statistical Analysis and Machine Learning Approaches

All analyses were implemented in Python (version 3.10), with complete computational workflows and reproducibility materials available in our public GitHub repository (https://github.com/arvidl/ibs-brain). Our analytical approach combined traditional statistical methods with advanced machine learning techniques, employing both parametric and non-parametric approaches as appropriate for the data distributions.

For group comparisons, statistical significance was assessed using a threshold of p<0.05, with Bonferroni corrections applied to control for multiple comparisons. Effect sizes were quantified using Cliff’s delta [37], a robust non-parametric measure particularly suitable for non-normally distributed data [38]. Following established conventions, we interpreted Cliff’s delta (absolute) values as negligible (0.00–0.14), small (0.15–0.33), medium (0.34–0.47), or large (0.48–1.00).

Relationships between variables were evaluated using Spearman’s rank correlation coefficient (ρ), chosen for its robustness to non-normality and ability to capture monotonic relationships [39]. Correlation strengths were classified as weak (0.20–0.39), moderate (0.40–0.59), strong (0.60–0.79), or very strong (0.80–1.00). Values below 0.20 were considered negligible to minimize the risk of over-interpreting weak associations.

To ensure reproducibility and transparency, all analysis scripts, including data preprocessing steps, statistical analyses, and visualization code, are documented in Jupyter notebooks accessible through our GitHub repository. These notebooks provide detailed documentation of parameter choices, statistical assumptions, and analytical decisions.

Our analysis strategy addressed four interconnected research objectives, progressing from replication to more advanced multivariate approaches, as set out below.

### 2.6. Research Objectives and Analytical Approach

**A** **—**
**Replication Analysis:**
Is it possible to replicate the morphometric findings of Skrobisz et al. [23] regarding IBS versus HC discrimination, using the same FreSurfer-derived features and the same FreeSurfer version?**(i)** By employing a feature-by-feature (univariate) comparison incorporating effect size?**(ii)** By employing a novel consistency score, combining several metrics for replication assessment?**B** **—**
**Software Version Comparison:**
Are there IBS versus HC disparities in morphometric feature values between FreeSurfer 6.0.1 and FreeSurfer 7.4.1 applied to the same set (n=78) of T1-weighted recordings in our Bergen cohort?What is the difference in the results between FreeSurfer 7.4.1 cross-sectional analysis versus FS 7.4.1 longitudinal stream?**(i)** When employing a feature-by-feature comparison?**(ii)** When employing a multivariate comparison, incorporating covariance structures in the morphometric features?**C** **—**
**Morphometric Classification Analysis:**
Is it possible to separate IBS individuals from HCs based on morphometric features?**(i)** By employing a feature-by-feature comparison (FS 7.4.1)?**(ii)** By employing a multivariate comparison, incorporating covariance structures in the morphometric features?**(iii)** By predicting IBS versus HC from the morphometric features using a machine learning framework (ML)?**(iv)** By identifying the importance of morphometric measures in the model with the best prediction?**D** **—**
**Integrated Morphometric–Cognitive Analysis:**
Would adding cognitive performance as a predictor improve the accuracy of separating IBS from HC?**(i)** By employing a feature-by-feature comparison?**(ii)** By employing a multivariate comparison, incorporating covariance structures in the cognitive features?**(iii)** By predicting IBS versus HC from morphometric and cognitive characteristics using a machine learning framework (ML)?**(iv)** By identifying the importance of morphometric and cognitive measures included in the model with the best prediction?

This hierarchical analytical framework progresses from basic replication to more advanced multivariate approaches, enabling both methodological validation and novel insights into IBS-related brain structure and function.

### 2.7. Statistical Analysis Framework

Given the complexity of our research questions and the combination of traditional and advanced analytical methods, we implemented a comprehensive statistical framework encompassing both univariate and multivariate approaches. Here, we detail our analytical strategy and its methodological justification.

### 2.8. Exploratory and Univariate Analyses

Initial analyses followed established protocols, as in [23], beginning with an exploratory data analysis of numerical features and cross-tabulation of categorical variables (Group: HC/IBS; sex: F/M). For univariate comparisons (Objectives A–D), we employed both parametric (independent *t*-tests) and non-parametric (Mann–Whitney U) tests, depending on normality assessments. Multiple comparison correction used the Bonferroni method, and effect sizes were quantified using Cohen’s d (for parametric tests) and Cliff’s delta [37] in all other cases. Cliff’s delta (δ) between two groups *X* and *Y* is defined as $\delta =\frac{U}{n_{x}n_{y}}-0.5$, where *U* is the Mann–Whitney U statistic, $n_{x}$ is the number of observations in group X, and $n_{y}$ is the number of observations in group Y. The resulting Cliff’s delta (δ) ranges from −1 to +1, where δ=+1 indicates that all values in group X are greater than all values in group Y, δ=−1 indicates that all values in group X are less than all values in group Y, and δ=0 indicates complete overlap between the two groups.

### 2.9. Permutation Testing

To address small sample sizes and potential non-normal distributions, we employed permutation testing (1000 iterations) to assess statistical significance. For each test, we computed an observed test statistic (sum of squared differences between group means) and generated a null distribution by randomly reassigning group labels. The empirical *p*-value was calculated as the proportion of permuted statistics exceeding the observed value. This non-parametric approach provides robust statistical inference while naturally controlling for multiple comparisons.

### 2.10. Multivariate Approaches—Assessing Multivariate Normality

For multivariate analyses (Objectives B–D), we first assessed multivariate normality using two complementary methods: Mardia’s test and the more comprehensive Henze–Zirkler’s test (see Section A.2 for details).

### 2.11. Advanced Distance Metrics

The Mahalanobis distance [40] quantifies the distance between a point *P* and a distribution *D* while accounting for data correlations [41]. Unlike Euclidean distance, it incorporates the covariance structure through the formula $D=\sqrt{{\left(x-\mu \right)}^{T}\;\Sigma^{-1}\;\left(x-\mu \right)}$, where *x* represents the data point, μ is the mean vector, and $\Sigma^{-1}$ is the inverse covariance matrix.

**Remark 1.** 
*While Cohen’s d ($d=\frac{\mu_{1}-\mu_{2}}{\sigma_{pooled}}$) measures standardized univariate group differences, the Mahalanobis distance extends this concept to multivariate space. In comparing IBS and HC groups, the squared Mahalanobis distance relates proportionally to Hotelling’s $T^{2}$ statistic, a multivariate analog of the squared t-statistic. Unlike Cohen’s d, which has standardized effect size interpretations (small: 0.2, medium: 0.5, large: 0.8), Mahalanobis distance interpretation depends on data dimensionality and covariance structure. To handle the outliers and non-normality common in neuroimaging data, we implemented a robust Mahalanobis distance. This modification employs winsorization (clipping values at 10th/90th percentiles) and replaces arithmetic means with medians (see Section A.3).*


### 2.12. Prediction of Group Belonging Using Machine Learning

In tasks **C(iii)** and **D(iii)**, we applied a comprehensive machine learning framework, utilizing morphometric features derived from FreeSurfer (aseg) to develop predictive models for two distinct classification tasks. We employed *PyCaret* version 3.3.2 (https://pycaret.org), an open-source, low-code machine learning library in Python, to develop and evaluate our classification models.

### 2.13. Machine Learning Model Development

Our machine learning approach followed a systematic protocol designed to ensure robust classification while addressing the challenges of limited sample size and potential overfitting. The analysis pipeline consisted of several carefully constructed stages optimized for neuroimaging data classification. Initial data preparation used a stratified sampling approach, partitioning the data set into training (70%) and testing (30%) sets while preserving the distribution of IBS/HC status across both partitions. This stratification was crucial for maintaining representative samples and ensuring valid model evaluation, particularly given our modest sample size and the inherent complexity of neuroimaging data. Model development utilized PyCaret’s comprehensive machine learning framework to evaluate multiple classification algorithms, ranging from traditional approaches to advanced ensemble methods. The classifier suite included linear models (logistic regression with L1 and L2 regularization), non-linear algorithms (support vector machines [SVMs] with various kernels), tree-based methods (random forests and gradient boosting machines, including XGBoost  [42] version 2.1.3 and LightGBM version 4.5.0), and instance-based learners (K-nearest neighbors). This diverse algorithm selection allowed exploration of different decision boundaries and pattern of feature interaction.

To ensure robust model assessment and mitigate overfitting risks, we implemented a nested 10-fold cross-validation strategy for model selection. This approach provided unbiased performance estimates while preventing data leakage between model selection and evaluation phases. The final model selection prioritized both predictive performance and model interpretability, considering the clinical relevance of our findings. See Table A1 for an illustration.

### 2.14. Model Performance Assessment

To address the class imbalance between IBS and HC groups, we implemented multiple complementary performance metrics. While classification *accuracy* served as a baseline measure, we employed additional metrics, such as the *F1 score* (harmonic mean of precision and recall) to balance false positive and negative rates; the receiver operating characteristic area under the curve (*ROC-AUC*) to assess discrimination ability across classification thresholds; and *Cohen’s Kappa* [43] to evaluate classification agreement beyond chance-level performance.

We generated *confusion matrices* to examine error patterns and potential classification biases. For analyses incorporating cognitive function, we used macro-averaged versions of these metrics, ensuring equal weighting across performance levels despite uneven class distributions. Performance assessment followed a dual-track strategy, evaluating models on both cross-validated training data and the held-out test set. This approach enabled us to assess both learning capacity and generalization ability, crucial considerations for clinical applications.

### 2.15. Feature Importance and Model Interpretability Analysis

To understand how morphometric and cognitive features contribute to classification performance, we implemented two complementary approaches to feature importance analysis: permutation importance and SHAP (SHapley Additive exPlanations) values.

The *permutation importance* [44] analysis quantifies feature relevance by measuring model performance degradation when individual features were randomly permuted. Through multiple iterations per feature, we calculated the mean decrease in model performance, providing a model-agnostic measure of feature importance.

The *SHAP analysis*, grounded in cooperative game theory [45], provided both global and local interpretation frameworks. The global analysis aggregated SHAP values across cases to identify consistently important features, while the local analysis examined feature contributions to individual predictions. We visualized these results using SHAP summary plots, which integrated both magnitude and directionality of feature effects.

By combining permutation importance with SHAP analysis, we gained complementary insights into feature relevance: permutation importance revealed features critical to overall model performance, while the SHAP analysis illuminated feature interactions and their contributions to specific predictions. This approach helped identify key neurobiological features distinguishing IBS patients from healthy controls, while exploring relationships between brain structure, sex differences, and cognitive function.

## 3. Results

### 3.1. Sample Demographics and Clinical Characteristics

The study enrolled 78 participants, comprising 49 patients with IBS and 29 HCs. Demographic analysis revealed comparable age distributions between groups (median age: IBS = 34 years, controls = 33 years). Female participants predominated in both cohorts, representing 77.6% (38/49) of the IBS group and 69.0% (20/29) of the control group, reflecting the typical gender distribution observed in IBS populations.

Symptom severity, quantified using the IBS Symptom Severity Scale (IBS-SSS), demonstrated clear differentiation between groups. The IBS cohort exhibited predominantly moderate to severe symptomatology, while healthy controls reported minimal gastrointestinal symptoms, aligning with our inclusion criteria. Six participants (three from each group) had missing IBS-SSS data, which we addressed through multiple imputations stratified by group and gender to maintain statistical robustness. Detailed demographic and clinical characteristics are presented in Table 2.

### 3.2. Replication Analysis of Skrobisz (2022) Using the Bergen Cohort (with FS 6.0.1)

In our Bergen cohort, we sought to replicate the morphometric findings reported by Skrobisz et al. [23] comparing IBS patients with healthy controls. Table 3 presents our comparative analysis using identical methodological parameters: 35 estimated total intracranial volume (eTIV)-normalized regional brain volumes derived from FreeSurfer 6.0, matching the analytical approach of the original study.

The volumetric comparison of brain structures between IBS patients and healthy controls across both cohorts reveals distinct patterns. While the Bergen cohort demonstrates systematically larger volumes (6–8% for global measures, reaching up to 35% for specific structures such as the *nucleus accumbens*), the within-cohort comparisons between IBS and healthy control groups show remarkable consistency in global brain eTIV-normalized volumes. Specifically, BrainSegVol values remain nearly identical within each cohort (Skrobisz: HC 0.753±0.018, IBS 0.749±0.016; Bergen: HC 0.805±0.025, IBS 0.806±0.024). Cortical measurements demonstrate similar stability, with total cortical volume (CortexVol) showing minimal between-group differences in both cohorts. In subcortical structures, we observed subtle variations, notably a slight trend toward volume reduction in IBS patients’ subcortical gray matter (SubCortGrayVol), though these differences remain within standard deviation bounds. White matter volumes maintain consistency between groups within cohorts, with an interesting pattern of white matter hypointensities emerging in the Bergen cohort. Corpus callosum segments exhibit relatively uniform volumes across all groups. Several methodological factors warrant consideration: the disparate cohort sizes (Skrobisz: HC n=19, IBS n=20; Bergen: HC n=29, IBS n=49), potential variations in FreeSurfer versions (6.0 versus 6.0.1), and differences in operating systems may contribute to the systematic volumetric differences observed between cohorts. While normalization to estimated total intracranial volume (eTIV) facilitates direct comparisons within cohorts by controlling for head size variation, it does not fully account for between-cohort differences.

Figure 3 presents a detailed reproducibility analysis, illustrating the differences in eTIV-normalized brain region volumes between HC and IBS across both cohorts. The plot contrasts effect sizes from the Skrobisz (2022) cohort (*x*-axis) against the Bergen cohort (*y*-axis), with the diagonal line representing perfect agreement. We employed Cohen’s d values for region-wise effect size calculations, as the availability of only parametric summary statistics from the Skrobisz study precluded non-parametric effect size measures. For each eTIV-normalized brain region volume and cohort, we calculated the pooled standard deviation as follows:$$s_{p}=\sqrt{\frac{\left(n_{1}-1\right)s_{1}^{2}+\left(n_{2}-1\right)s_{2}^{2}}{n_{1}+n_{2}-2}}$$
where $n_{1}$ and $n_{2}$ are the sample sizes and $s_{1}$ and $s_{2}$ are the standard deviations of the two groups, IBS and HC, respectively. Cohen’s d effect size was then computed as follows:$$d=\frac{{\bar{x}}_{1}-{\bar{x}}_{2}}{s_{p}}$$
where ${\bar{x}}_{1}$ and ${\bar{x}}_{2}$ are the means of the two groups. The 95% confidence interval for d was calculated using the following equation:$$CI_{95\%}=d\pm 1.96\sqrt{\frac{n_{1}+n_{2}}{n_{1}n_{2}}+\frac{d^{2}}{2(n_{1}+n_{2})}}$$
where the standard error term accounts for both sampling variance and uncertainty in the effect size estimate.

An *overall reproducibility score* (*S*) was developed for each brain region to quantify cross-cohort consistency through the following three complementary metrics: directional consistency (σ), confidence interval overlap (ω), and effect magnitude (ϵ). The score is computed as: S=σ+ω+ϵ, where the binary indicator σ equals one if the direction of effect is consistent between cohorts and equals zero otherwise; the binary indicator ω equals one if the 95% confidence intervals overlap and equals zero otherwise; and ϵ represents the minimum absolute effect size observed across cohorts.

This composite metric prioritizes brain regions exhibiting robust cross-cohort replication, with ϵ providing additional weight to stronger effects. Higher scores (*S*) indicate a greater reproducibility of morphometric findings across independent study populations and analysis pipelines, thereby establishing a quantitative framework for identifying the most reliable neuroanatomical alterations in IBS.

The effect size comparison between cohorts revealed moderate correlation (r = 0.203, *p* = 0.243). Directional consistency analysis demonstrated that 51.4% of brain regions maintained consistent IBS versus HC differences across cohorts. Notably, all brain regions exhibited overlapping 95% confidence intervals between cohorts, indicating that, despite differences in point estimates, the between-cohort variations did not reach statistical significance given measurement uncertainty. Five regions demonstrated particularly strong cross-cohort consistency, achieving the highest overall reproducibility scores (*S*), as follows: mid-anterior corpus callosum (CC Mid Anterior), Left Pallidum, Left Thalamus, Right Pallidum, and Left Amygdala. These structures showed overall scores ranging from 2.14 to 2.26, suggesting robust replication of IBS-related alterations. Conversely, several regions exhibited marked between-cohort divergence. White matter hypointensities demonstrated particularly discordant effects, while specific corpus callosum segments (CC Posterior and CC Mid Posterior) showed stronger effects in the Bergen cohort. Cerebellar regions clustered near the origin, indicating consistently modest effects across both cohorts. The overall pattern suggests limited agreement between cohorts in IBS-related brain alterations. While specific structures show robust reproducibility, the widespread dispersion around the diagonal reference line, coupled with moderate correlation, indicates substantial heterogeneity in morphometric findings between these independent samples. This variability may reflect genuine biological heterogeneity in IBS-related brain alterations or methodological differences between studies.

Figure 4 plots a ranking of brain regions on how consistently they show similar patterns between the cohorts.

The reproducibility analysis revealed varying degrees of cross-cohort consistency in brain structural alterations associated with IBS. Several regions demonstrated robust reproducibility, with the Left Pallidum, Left Thalamus, and CC Mid Anterior achieving overall scores (*S*) exceeding 2.0. These high-scoring regions exhibited both directional consistency and complete confidence interval overlap, coupled with substantial effect magnitudes, suggesting reliable IBS-related volumetric alterations across independent samples. Conversely, regions including the Right Caudate, Right Cerebellum Cortex, Left Hippocampus, Right Hippocampus, CC Mid Posterior, and Left Cerebellum Cortex showed lower reproducibility (scores of approximately 1.1). While these regions maintained confidence interval overlap, they lacked directional consistency between cohorts, suggesting greater variability in IBS-related effects. Despite systematic between-cohort differences in eTIV-normalized volumes, certain regions demonstrated consistent relative patterns of alteration. However, our attempt to replicate the specific morphometric differences reported by Skrobisz (2022) [23] yielded limited success. This suggests that structural brain alterations in IBS may be more heterogeneous than previously recognized, potentially reflecting the complex nature of IBS pathophysiology or methodological variations across studies.

To assess the robustness of brain morphometry measurements in IBS research, we conducted a comprehensive analysis of the Bergen cohort data using multiple FreeSurfer processing pipelines. This systematic evaluation examined the stability of morphometric measurements and IBS versus healthy control (HC) group differences across different analytical approaches: FreeSurfer versions (6.0.1 versus 7.4.1) and processing streams within FreeSurfer 7.4.1 (cross-sectional versus longitudinal). Our interventional study design enabled the application of the longitudinal processing stream, providing an additional dimension for assessing measurement reliability. Unlike our previous replication analysis of the [23] cohort, which relied on summary statistics, this comparison utilized complete morphometric data from all participants, allowing for more detailed assessment of measurement consistency.

### 3.3. Cross-Version Comparison of FreeSurfer Morphometric Measurements

We examined the consistency of volumetric measurements between FreeSurfer versions 6.0.1 and 7.4.1 (cross-sectional stream) in quantifying brain structural differences between IBS patients and healthy controls (HC). Table A2 in the Section A.4 presents group-wise summary statistics (mean and standard deviation) for both IBS patients and healthy controls, derived from the aseg.stats files generated by each FreeSurfer version. Figure 5 presents a scatter plot matrix illustrating version-wise comparisons for each brain region. Individual plots display FS 6.0.1 volumes against corresponding FS 7.4.1 measurements, with HC and IBS participants distinguished by blue and red markers, respectively. Reference identity lines facilitate the direct assessment of cross-version measurement concordance.

The scatter plot matrix demonstrates varying degrees of consistency between FreeSurfer versions 6.0.1 and 7.4.1 across different brain regions. Subcortical regions, particularly the thalamus, caudate, putamen, and partly the hippocampus, show strong cross-version agreement with minimal deviation from the identity line. However, systematic differences emerge in several structures: the amygdala and the accumbens demonstrate moderate version-dependent variability, with data points showing systematic deviation from perfect concordance. Corpus callosum segments display region-specific variations in cross-version agreement, with CC Anterior and CC Mid Anterior showing more pronounced differences compared to other segments. Importantly, the distribution patterns of IBS (red) and healthy control (blue) groups remain fairly consistent across versions, suggesting that, while absolute volume estimates may differ between FreeSurfer versions, the relative group differences are largely preserved.

Notably, several regions exhibit strong correlations between versions but with systematic offsets from the identity line, indicating consistent biases between FreeSurfer versions 6.0.1 and 7.4.1. For example, the cortical measurements (lhCortexVol and rhCortexVol), lh- and rhCerebralWhiteMatterVol, and TotalGrayVol show a clear parallel offset above the identity line, indicating that FreeSurfer 6.0.1 consistently produces higher volume estimates than version 7.4.1. This systematic bias appears consistent across the full range of eTIV-normalized volumes and both subject groups. Similar parallel offsets are visible in Left Cerebellum Cortex and Right Cerebellum Cortex and in subcortical structures like the Left Pallidum and Left Caudate. Moreover, the eTIV shows systematic higher volumes in version 7.4.1 than in version 6.0.1 measurements.

Several key structures exhibit individual outliers that warrant attention. In eTIV, a single measurement shows substantial deviation, suggesting potential segmentation challenges in this particular case. The Left Hippocampus and Right Hippocampus both show isolated outliers (visible as blue points) significantly deviating from the otherwise tight correlation pattern, indicating potential segmentation inconsistencies between versions for these specific control subjects. The Left Thalamus displays a particularly notable outlier (blue point) that deviates substantially below the main correlation pattern, suggesting a case where version 7.4.1 produced a markedly lower volume estimate compared to version 6.0.1. Similar isolated discrepancies appear in both Left Amygdala and Right Amygdala measurements, where single data points (again from the control group) deviate notably from the otherwise consistent version correlation. These individual outliers likely represent cases where the segmentation algorithms in the two FreeSurfer versions interpreted the anatomical boundaries differently, possibly due to image quality issues, anatomical variants, or differences in how the versions handle boundary cases. The fact that many of these outliers appear in the control group (blue points) suggests that these discrepancies are not specifically related to IBS pathology but rather to technical aspects of the segmentation process.

These observations underscore the importance of version consistency in morphometry-based classification studies and suggest that meta-analyses or multi-site studies should carefully account for FreeSurfer version effects in their analytical pipelines.

In this context, Figure 6 depicts a scatter plot matrix comparing brain region volumes between two pipelines (cross-sectional and the longitudinal stream) using the *same* FreeSurfer 7.4.1 version, highlighting potential discrepancies.

The comparison between FreeSurfer 7.4.1’s cross-sectional and longitudinal processing streams reveals distinct patterns of agreement and systematic variation across brain regions. Global measurements (e.g., BrainSegVol, BrainSegVolNotVent) demonstrate strong cross-stream consistency, with tight clustering along the identity line. However, substantial systematic differences emerge in several key structures. Most notably, cortical volumes (i.e., lhCortexVol, rhCortexVol) exhibit a clear systematic bias, with longitudinal processing consistently producing higher volume estimates than the cross-sectional stream. This pattern contrasts with the Left Cerebellum Cortex and Right Cerebellum Cortex, where longitudinal processing yields systematically lower estimates. Subcortical structures display varying degrees of processing stream sensitivity: the putamen and caudate show consistent offsets from the identity line, while pallidum and accumbens measurements demonstrate greater scatter. Corpus callosum segments (CC Anterior, CC Mid Anterior, and CC Central) reveal processing stream-dependent variations that differ from those observed in other structures. Looking at the eTIV plot in the top-left panel, it shows remarkably high consistency between cross-sectional and longitudinal processing streams. The data points cluster tightly along the identity line across the full range of values (approximately 1.2–1.8 × 10^6^ mm^3^), with minimal deviation. This strong agreement in eTIV estimations between processing streams is particularly noteworthy because eTIV serves as the normalization factor for all other volumetric measurements. The consistency suggests that any observed differences in other brain regions are not attributable to variations in total intracranial volume estimation between processing streams, but rather reflect genuine methodological differences in how the two streams segment specific structures.

Importantly, these systematic biases maintain consistency across both IBS and healthy control groups, as evidenced by the parallel patterns of red and blue markers. This indicates that, while absolute volume estimates differ between processing streams, the relative group differences remain largely preserved. These findings underscore the critical importance of maintaining consistent processing stream selection when conducting cross-sectional comparisons or longitudinal analyses in clinical studies.

The summary statistics, specifically the means and standard deviations for the Freesurfer v. 7.4.1 cross-sectional and the v. 7.4.1 longitudinal stream, respectively, are shown in the Section A.5 in Table A3.

Figure 7 illustrates the differential impact of FreeSurfer processing choices on IBS versus healthy control effect sizes across brain regions. Panel (a) compares effect sizes between FreeSurfer versions 6.0.1 and 7.4.1 (cross-sectional), while panel (b) contrasts effect sizes derived from FreeSurfer 7.4.1’s cross-sectional and longitudinal processing streams, enabling the assessment of both version and pipeline-specific influences on group differences.

The scatter plots reveal distinct patterns in how FreeSurfer methodological choices affect IBS versus healthy control effect sizes across brain regions. Panel (a), comparing FreeSurfer versions 6.0.1 and 7.4.1 (cross-sectional), demonstrates moderate agreement with notable version-specific variations. Key corpus callosum segments (CC Anterior, CC Mid Posterior) show the strongest positive effect sizes (approximately 0.4) and maintain relative consistency across versions. In contrast, the Left Accumbens Area exhibits the strongest negative effect (approximately −0.4), with its magnitude varying between versions. Panel (b), comparing cross-sectional and longitudinal streams within FreeSurfer 7.4.1, shows that corpus callosum segments maintain their position as regions with the strongest positive effects, while the Left Amygdala and Left Accumbens Area show pronounced negative effects. Most subcortical structures cluster more tightly around the diagonal compared to the version comparison in panel (a). The longitudinal versus cross-sectional comparison demonstrates greater overall consistency than the version comparison, as evidenced by tighter clustering along the diagonal reference line. This suggests that processing stream selection within FreeSurfer 7.4.1 introduces less variability in effect size estimates than version changes. However, specific regions, particularly in the limbic system, show sensitivity to processing stream choice. This systematic comparison highlights that, while both FreeSurfer version and processing stream selection affect effect size estimates, version differences generally introduce more variability than processing stream choices within the same version.

Figure 8 quantifies the reproducibility of IBS versus healthy control group differences across brain regions under different FreeSurfer methodological variants. Panel (a) ranks regions by their effect size consistency (*S*) between FreeSurfer versions 6.0.1 and 7.4.1 (cross-sectional), while panel (b) presents regional rankings based on effect size stability between cross-sectional and longitudinal processing streams within FreeSurfer 7.4.1, enabling the systematic assessment of both version- and pipeline-dependent variations.

The regional consistency scores reveal distinct patterns in how FreeSurfer methodological choices affect the reproducibility of IBS versus healthy control differences. Panel (a), comparing FreeSurfer versions 6.0.1 and 7.4.1 (cross-sectional), shows a gradual distribution of consistency scores ranging from 1.0 to 2.5. Corpus callosum regions (CC Mid Posterior, CC Posterior) demonstrate the highest consistency, while cerebellar structures show the lowest. Subcortical regions exhibit intermediate consistency, suggesting moderate stability across FreeSurfer versions. Panel (b), comparing cross-sectional and longitudinal streams within FreeSurfer 7.4.1, reveals a more distinct clustering pattern. The CC Anterior and CC Mid Posterior maintain high consistency, but, notably, limbic structures like the Left Amygdala and Right Thalamus show improved consistency compared to their version-wise rankings. This suggests that these regions are more sensitive to FreeSurfer version changes than to processing stream selection. The overall pattern indicates stronger methodological stability when varying processing streams within FreeSurfer 7.4.1 compared to cross-version analyses. Importantly, comparing these methodological variations within the same cohort yields higher consistency scores than the previous cross-cohort comparison (Figure 4), highlighting the substantial impact of cohort-specific factors on brain morphometric findings in IBS research.

### 3.4. Multivariate Analyses: IBS Versus HC

The multivariate normality of brain structural data was assessed across three FreeSurfer processing streams using Mardia’s test (examining skewness and kurtosis) and Henze–Zirkler’s test. For FS 6.0.1, Mardia’s test revealed significant deviations in both skewness ($b_{1,p}=2.33\times 10^{14}$, p<0.001) and kurtosis ($b_{2,p}=-8.77$, p<0.001) for the full sample, with similar patterns in the IBS group but different skewness characteristics in the HC group. For the FS 7.4.1 cross-sectional stream, both groups showed significant non-normality, with particularly extreme values in the IBS group (kurtosis statistic = 153.63, p<0.001). The FS 7.4.1 longitudinal analysis also indicated significant departures from multivariate normality across all groups. The Henze–Zirkler’s test showed some numerical instability issues, evidenced by extreme values and negative test statistics, suggesting that its results should be interpreted with caution. Overall, these findings consistently indicate significant departures from multivariate normality across all FreeSurfer versions and subject groups, with particularly pronounced effects in the IBS group. This suggests that robust statistical methods should be employed for subsequent analyses of group differences in brain structure.

In this context, the robust Mahalanobis distance analysis was implemented to quantify the multivariate separation between IBS and HC groups across different FreeSurfer processing streams while accounting for potential outliers and non-normality in the neuroimaging data. The computation employs winsorization at the 10th and 90th percentiles to mitigate the impact of extreme values, followed by robust location estimation using medians instead of means. The analysis revealed the effects of decreasing Mahalanobis distances across FreeSurfer versions: FS 6.0.1 showed the largest separation (D=9.348, F=0.598, p=0.939), followed by FS 7.4.1 cross-sectional (D=6.068, F=0.252, p≈1.000) and FS 7.4.1 longitudinal (D=5.163, F=0.183, p≈1.000). However, none of these distances reached statistical significance (all p>0.05), suggesting that the multivariate brain volume differences between IBS and HC groups are not statistically meaningful across any of the FreeSurfer processing streams. The consistently high *p*-values and low F-statistics indicate that, despite the apparent numerical differences in Mahalanobis distances, there is insufficient evidence to conclude that the IBS and HC groups differ significantly in their multivariate brain volume profiles. This analysis, incorporating 35 brain regions and accounting for their covariance structure, suggests that the volumetric differences between IBS and HC groups are not robust enough to clearly distinguish between the groups in a multivariate framework.

To further investigate potential group differences beyond the initial Mahalanobis distance analysis, we employed a machine learning framework with cross-validation to assess IBS versus healthy control discriminability and identify the most diagnostically relevant brain structures. This complementary approach enables systematic evaluation of multivariate patterns while accounting for potential interactions between brain regions.

### 3.5. Machine Learning-Based Classification Using Brain Morphometry

We evaluated the discriminative power of brain morphometric features for IBS versus healthy control classification using the PyCaret machine learning library. Multiple classification algorithms were trained and compared (Figure A1) using FreeSurfer 7.4.1 longitudinal stream measurements from the Bergen cohort (Table 2). We applied a binary classification framework to distinguish between healthy controls (0) and IBS patients (1) based on brain morphometric features. The dataset comprised 78 participants, characterized by 37 numerical features, who were divided into a training set (n = 54) and a test set (n = 24). We employed stratified 10-fold cross-validation to maintain consistent class proportions across folds. Feature preprocessing included mean-based imputation and standardization to zero mean and unit variance, particularly crucial for features with widely differing scales (e.g., raw eTIV values $>1.2\cdot 10^{6}$ versus eTIV-normalized measures <1). Given the modest dataset size, analyses were performed using CPU computation. All random processes were controlled through a fixed session identifier to ensure reproducibility.

Model performance evaluation across 15 classification algorithms revealed extreme gradient boosting (XGBoost) as the superior approach for IBS versus healthy control discrimination based on brain morphometry (details in Figure A1). XGBoost achieved the highest scores on the following performance metrics: accuracy (0.72), AUC (0.68), recall (0.72), precision (0.74), and F1 score (0.71). The model’s Cohen’s kappa (0.40) and Matthews correlation coefficient (0.42) indicated substantial improvement over chance-level classification. K-nearest neighbors demonstrated the second-best performance, while logistic regression and support vector machines showed moderate discriminative ability. Several algorithms, including AdaBoost and linear discriminant analysis, performed near chance level, as benchmarked against a dummy classifier baseline. XGBoost’s superior performance suggests its ability to capture complex, nonlinear relationships in brain morphometric features that distinguish IBS from healthy controls.

The best-performing model (XGBoost) demonstrated mixed classification performance on the hold-out test set, as shown in Figure 9a. The model correctly identified 73% of IBS patients (eleven of fifteen cases; eight females, three males; IBS-SSS: 245.7 ± 60.4; age: 33.2 ± 7.6). However, specificity was low at 11%, with eight of nine healthy controls misclassified as IBS (three females, five males; IBS-SSS: 19.2 ± 19.6; age: 25.4 ± 5.7), yielding an overall accuracy of 50% (12/24). This asymmetric performance reveals systematic patterns: correctly classified IBS patients showed higher symptom severity scores (IBS-SSS), female predominance, and a higher mean age compared to misclassified controls. The strong bias toward IBS classification suggests that, while brain morphometric features contain discriminative information, additional refinement is needed for reliable diagnostic application.

The permutation importance analysis revealed the relative contribution of brain regions to IBS versus healthy control classification. The central corpus callosum (CC Central) emerged as the most discriminative feature (≈0.057±0.038), followed by white matter hypointensities (≈0.043±0.029) and the left nucleus accumbens (≈0.035±0.040). A second tier of discriminative regions includes the mid-posterior corpus callosum (≈0.033±0.029) and left amygdala (≈0.028±0.045), while cerebellar structures showed moderate importance (right cerebellar cortex (≈0.022±0.026). Notably, several traditionally studied regions in IBS, including the hippocampus (≈0.014±0.045) and total intracranial volume (≈0.015±0.028), demonstrated relatively lower discriminative power. This hierarchy suggests that white matter structures, particularly corpus callosum segments, may play a more prominent role in IBS-related brain alterations than previously recognized. However, the permutation importance ranking should be interpreted cautiously, given the large standard deviations and the model’s modest classification performance (50% accuracy and 73% sensitivity but only 11% specificity). Additionally, while the ranking identifies features that contribute most to the model’s decisions, these contributions come from a model that shows strong bias toward IBS classification and poor discriminative ability for healthy controls.

To gain deeper insight into how individual brain regions influence the model’s classification decisions, we employed SHAP (SHapley Additive exPlanations) analysis. Figure 10 visualizes the contribution of each morphometric feature to individual predictions, with SHAP values indicating both the direction and magnitude of each feature’s impact. This analysis extends beyond traditional feature importance rankings by revealing how specific volumetric measurements drive classification outcomes on a case-by-case basis. High feature values (red) and low feature values (blue) can contribute differently to the model’s decisions, providing a more nuanced understanding of the relationship between brain morphometry and IBS classification than permutation importance alone. The figure reveals complex patterns in how morphometric features influence predictions. For example, high values (red) in the right caudate tend to push predictions toward IBS (positive SHAP values), while low values (blue) in this region tend to predict healthy control. This asymmetric impact of feature values suggests nonlinear relationships between brain structure volumes and IBS classification that may not be captured using simpler univariate analyses.

The SHAP analysis reveals more nuanced feature contributions than the permutation importance ranking, while also showing some notable consistencies. CC Central ranks highest in permutation importance and shows meaningful SHAP values, but with complex patterns where both high and low values contribute to classification. Similarly, CC Mid Posterior shows similar importance in both analyses, with relatively consistent effects. White matter features, particularly WM Hypointensities, rank high in both analyses, suggesting robust importance, with SHAP patterns indicating that higher values tend to predict healthy controls. Among subcortical structures, the Left Accumbens Area appears important in both analyses, with SHAP values showing that lower volumes tend to predict IBS. The Left Amygdala shows moderate importance in both analyses, with high values generally predicting healthy controls. Notable differences emerge: the Right Caudate shows strong SHAP value patterns but does not appear in the top permutation importance features, while the Right Hippocampus ranks lower in permutation importance but shows distinct SHAP patterns. This comparison suggests that, while some features (like corpus callosum regions and white matter hypointensities) show consistent importance across methods, the SHAP analysis reveals more complex relationships between feature values and model predictions. This richer characterization of feature contributions might explain some of the model’s classification biases, particularly given the observed asymmetric effects where high and low values of the same feature can have different impacts on predictions. However, these feature contribution analyses must (again) be interpreted in the context of the model’s modest classification performance (50% accuracy, 73% sensitivity, and 11% specificity). The SHAP values and permutation importance rankings identify features that drive the model’s decisions, but given the strong bias toward IBS classification, these patterns may reflect systematic misclassification rather than truly discriminative neuroanatomical markers. The complex feature interactions revealed by SHAP analysis might partially explain the model’s poor specificity, suggesting that while consistent morphometric patterns exist, they are insufficient for reliable diagnostic classification without additional clinical information.

### 3.6. Univariate Analysis of Cognitive Performance

To assess potential cognitive differences between IBS patients and healthy controls, we analyzed performance across multiple cognitive domains using the Repeatable Battery for the Assessment of Neuropsychological Status (RBANS). Table 4 presents group comparisons of the full-scale score and five cognitive domain indices, using non-parametric statistics to account for potential non-normal distributions.

The comparison of cognitive performance between IBS patients and healthy controls using RBANS revealed domain-specific differences. Following Bonferroni corrections (α=0.05/6), two measures showed significant group differences: the full-scale RBANS score was lower in IBS patients compared to controls, with a small to moderate effect size. Similarly, the recall index demonstrated significantly lower performance in IBS patients compared to controls. Other cognitive domains showed no significant differences after correction, including memory index, visuospatial index, verbal skills index, and attention index. These findings suggest selective cognitive differences in IBS, particularly affecting overall cognitive function and recall abilities, while other domains remain relatively preserved. The use of Bonferroni correction provides strong control of the Type I error false positive) rate in these multiple comparisons, though its conservative nature may increase the risk of Type II errors (failing to detect true differences).

### 3.7. Relationship Between Brain Morphometry and Cognitive Performance

To investigate potential links between brain structure and cognitive function, we examined pairwise correlations between regional brain volumes and RBANS cognitive scores. Figure 11 presents a correlation matrix using Spearman’s rank correlation, capturing both linear and nonlinear monotonic relationships while remaining robust to outliers and non-normal distributions. This comprehensive analysis includes both morphometric features (regional volumes normalized by eTIV) and cognitive performance measures across multiple domains.

The correlation analysis reveals three distinct patterns of relationships. First, strong bilateral symmetry is evident in subcortical structures, with high correlations between corresponding left and right regions (hippocampus: ρ≈0.8, amygdala: ρ≈0.7, putamen: ρ≈0.9). Second, anatomical relationships appear preserved, with TotalGrayVol showing expected moderate correlations with subcortical structures (ρ≈ 0.4–0.6) and corpus callosum segments displaying varying degrees of inter-relationship (ρ≈ 0.2–0.7). However, the structure–function relationships, as measured by correlations between brain morphometry and cognitive performance, are notably weak. The Full-scale RBANS shows minimal correlations with regional volumes (|ρ|<0.25), and even theoretically linked relationships, such as between Memory Index and medial temporal structures, demonstrate weak associations (|ρ|<0.15). An unexpected finding is the moderate correlation between WM Hypointensities and Visuospatial Index (ρ=0.33), while other structure–function correlations remain weak (|ρ|<0.15). Within cognitive measures, moderate to strong inter-correlations exist among most RBANS indices, particularly between memory-related measures (Recall Index and Memory Index: ρ=0.67), suggesting preserved cognitive domain relationships despite weak associations with brain structure. This pattern indicates that the relationship between brain morphometry and cognitive function in IBS may be more complex than direct structure–function mappings would suggest.

### 3.8. Multimodal Classification of IBS Using Brain Structure and Cognitive Measures

To evaluate whether combining brain morphometry with cognitive performance improves diagnostic classification, we implemented machine learning models using both feature types. We systematically compared classification performance between models trained on morphometric features alone versus those incorporating both morphometric and cognitive measures. Figure 12a presents the detailed classification outcomes, while Figure 12b shows the relative importance of combined features in the model’s decision-making. Table 5 quantifies the impact of feature combination through multiple performance metrics. Results are shown for the XGBoost model (which was ranked second best, after knn).

The confusion matrix in Figure 12a illustrates the XGBoost model’s classification performance using combined brain morphometry and cognitive features. The model demonstrates high sensitivity but poor specificity in IBS detection. Among IBS patients, 14 of 15 were correctly identified (93.3% sensitivity), with these true positives showing characteristic IBS-SSS scores (271.3±81.0) and female predominance (11F/3M). However, the specificity was low (22.2%), with only two of nine healthy controls correctly classified. The misclassification patterns reveal notable demographic and clinical features. The false positives (seven controls misclassified as IBS) show a male predominance (5M/2F) and lower age (25.7±6.1 years) compared to true positives, despite normal IBS-SSS scores (19.6±21.2). The single false negative case presents distinct characteristics: male, older (47.0 years), with substantial symptom severity (IBS-SSS: 245.0). These classification outcomes suggest that, while the combined morphometric and cognitive features enable sensitive IBS detection, they lack specificity. The gender-specific misclassification patterns and age-related differences in classification accuracy indicate potential demographic influences on the model’s performance. These findings highlight both the promise and limitations of multimodal classification approaches in IBS diagnosis.

Feature importance analysis (Figure 12b) reveals the relative contributions of brain structural and cognitive measures to IBS classification. The right hippocampus emerges as the most discriminative feature (importance ≈0.073±0.036), followed by the right pallidum and left cerebellar white matter (with importance scores of ≈0.032±0.026, and ≈0.031±0.029, respectively). Notably, cognitive performance, represented by the Recall Index and Verbal Skills Index, ranks among the top discriminative features, suggesting that the integration of cognitive measures enhances classification performance. The ranking highlights a mixed contribution of structural and cognitive features, with subcortical structures (Right Hippocampus, Right Pallidum, Left Accumbens Area) showing particularly strong discriminative power. Global brain measures (CortexVol, BrainSegVol, BrainSegVolNotVent) demonstrate minimal importance, suggesting that regional rather than global alterations better distinguish IBS from healthy controls. This importance ranking should be interpreted in the context of the model’s classification performance metrics, where, despite improved sensitivity with combined features, specificity remains low. The prominence of memory-related structures and cognitive measures aligns with the observed group differences in RBANS scores, providing a potential neurobiological basis for cognitive alterations in IBS.

Table 5 quantifies the impact of incorporating cognitive measures into the morphometry-based classification through comprehensive performance metrics. The addition of cognitive features to brain morphometry (M ∪ C) substantially improved model performance across multiple dimensions: sensitivity increased from 73.3% to 93.3%, accuracy from 50.0% to 66.7%, and the F1 score from 0.647 to 0.778; also, while specificity remained modest, it showed improvement from 11.1% to 22.2%. The Matthews correlation coefficient (MCC) shifted from −0.185 to 0.228, indicating an enhanced overall classification performance when combining both feature types.

SHAP analysis (Figure 13) reveals the complex interactions between brain structure, cognitive performance, and IBS classification. The right hippocampus demonstrates the strongest feature impact, with higher volumes (red) generally predicting healthy control status and lower volumes (blue) predicting IBS. The Verbal Skills Index emerges as the second most influential feature, showing a distinct pattern where lower scores tend to predict IBS classification. Among subcortical structures, the right caudate and putamen show notable but contrasting patterns. The right caudate exhibits a clustered distribution with clear value-dependent effects, while the right putamen shows more dispersed impact across participants. The left cerebellar white matter demonstrates moderate influence, with its effect direction varying based on volume. The overall pattern suggests a hierarchical organization of discriminative features, where both structural and cognitive measures contribute to classification decisions. Lower-ranked features, including global measures (TotalGrayVol, lhCortexVol) and white matter hypointensities, show minimal impact on model predictions, suggesting that regional rather than global alterations better characterize IBS-related brain differences.

## 4. Discussion

Our integrated analysis of brain structure and cognitive function in IBS yields several principal findings with important methodological and clinical implications. First, despite using identical FreeSurfer versions and processing pipelines, we were unable to replicate the morphometric differences between IBS and healthy controls reported by Skrobisz et al. (2022). Specifically, while they found reduced thalamic volume in IBS patients, our analysis showed no significant volumetric differences in this or other brain regions. This non-replication warrants particular consideration—it may reflect true biological heterogeneity in IBS, differences in patient characteristics between cohorts (however, both cohorts utilized the standardized Rome IV diagnostic IBS criteria), or highlight critical methodological sensitivities in brain morphometry studies.

The systematic comparison of FreeSurfer versions 6.0.1 and 7.4.1 revealed substantial version-dependent variations in morphometric measurements. Global brain volumes demonstrated systematic offsets (6–8%), with even larger discrepancies (up to 35%) in specific structures like the nucleus accumbens, while relative group differences were largely preserved across versions, absolute measurements showed consistent biases, particularly in subcortical and limbic regions. For example, cortical measurements (lhCortexVol, rhCortexVol) exhibited parallel offsets above the identity line, indicating that FreeSurfer 6.0.1 consistently produced higher volume estimates compared to version 7.4.1. Similar systematic biases were observed in cerebellar structures and several subcortical regions. In this context, we will also mention our contribution to reproducibility analysis. For region-wise comparisons, we have introduced the concept and definition of a *reproducibility score*, *S* (=σ+ω+ϵ), to quantify cross-cohort consistency through the following three complementary metrics: directional consistency (σ), confidence interval overlap (ω), and effect magnitude (ϵ).

Our machine learning analyses reveal a complex relationship between brain structure, cognitive function, and IBS classification, while morphometric features alone showed limited discriminative power (sensitivity 73.3%, specificity 11.1%), the *integration of cognitive measures substantially improved classification accuracy*. The combined model achieved a notably higher sensitivity (93.3%), correctly identifying 14 of 15 IBS patients, with these true positives showing characteristic IBS-SSS scores (271.3 ± 81.0) and female predominance (11F/3M). However, specificity remained modest (22.2%), with only one of nine healthy controls correctly classified. This asymmetric performance pattern, particularly the high false positive rate among male controls (5M/2F, age 25.7 ± 6.1 years), suggests that, while brain structural and cognitive alterations may be characteristic of IBS, they are not necessarily specific to the condition.

Feature importance analysis provides insight into this classification pattern. The right hippocampus emerged as the most discriminative feature (importance ≈0.073±0.036), followed by subcortical structures (right pallidum, left cerebellar white matter) and cognitive measures (Recall Index, Verbal Skills Index). This hierarchy, consistently identified through both permutation importance and SHAP analyses, suggests a fundamental relationship between memory-related neural circuits and IBS pathophysiology. The prominence of hippocampal measurements aligns with emerging evidence for altered brain-gut-behavior interactions in IBS [14], particularly regarding the role of memory systems in visceral symptom processing and learned pain responses [46,47].

### 4.1. Brain Structures Involved in Discriminating Between IBS and HC

Our results showed that subcortical structures, particularly within the basal ganglia (caudate, putamen, pallidum), played an important role in distinguishing IBS patients from healthy controls. While traditionally associated with motor control, the basal ganglia also critically influences reward processing, habit formation, and pain modulation—functions that are directly relevant to IBS symptomatology and its impact on patients’ experience of gastrointestinal symptoms. These findings align with recent results from a UK Biobank study [20], which also highlighted the importance of hippocampal and basal ganglia structures, including the pallidum and caudate, in IBS.

Beyond the basal ganglia, several other subcortical structures relevant to IBS symptomatology emerged as discriminators. The nucleus accumbens, fundamental to reward processing and motivation, may mediate the emotional and motivational aspects of chronic pain in IBS. Dysfunction in this structure could explain the intensified emotional distress and pain sensitivity commonly reported by IBS patients [9]. Similarly, the amygdala appears significant, particularly given its connection to pain modulation and emotion-processing networks, including the prefrontal cortex and insula. This aligns with previous research [48] demonstrating enhanced amygdala–insula connectivity in IBS patients. Although our results differ from Skrobisz et al.’s [23] findings regarding thalamic involvement, other studies have supported its role in IBS. Diffusion tensor imaging has revealed altered thalamic organization in IBS patients, with reduced fractional anisotropy and increased mean diffusivity [49]. These alterations suggest compromised structural integrity of thalamic circuits, potentially affecting pain processing and sensory integration. The involvement of corpus callosum should also be mentioned, as interhemispheric integration is crucial for visceral sensation processing and pain modulation [50], as well as in mental disorders [51]. Taken together, our findings support integrated neural signatures being involved in predicting IBS [52].

### 4.2. Integration of Cognitive Performance and Brain Structure in IBS

The enhanced diagnostic accuracy due to including cognitive measures strongly supports that IBS should be understood as a disorder of the gut–brain interaction [14,53]. The brain’s integral role in cognitive, emotional, and autonomic regulation suggests that these manifestations are fundamentally interconnected rather than merely coincidental. The prominent role of hippocampal volume was a principal finding. The fundamental role of the hippocampus in cognitive processing is well known [54], and was supported by the Recall Index being identified as another feature with strong importance. The role of verbal skills was more surprising. Although research has established connections between memory systems and language processing, particularly in semantic memory organization [55], a negligible correlation between the two indices suggests that IBS affects multiple cognitive domains through independent mechanisms.

Our findings may also have implications for other somatic and psychiatric disorders, like Alzheimer’s disease, Parkinson’s disease, and major depression. The gut–brain axis is involved in all these diseases, which also are characterized by cognitive impairment. Recent research has identified potential pathways linking gut microbiota alterations to neurological function, particularly through inflammatory responses and tryptophan metabolism [56,57]. The emergence of the microbiota–gut–brain axis as a key framework [58] offers new perspectives on how peripheral inflammation might influence both brain structure and cognitive function in IBS. This integrated view suggests that cognitive assessment, combined with brain morphometry, might provide valuable insights not only for IBS but for a broader spectrum of gut–brain disorders.

### 4.3. Brain–Gut Axis: Implications for Understanding and Treating IBS

Our findings should have important implications for clinical practice and treatment strategy. The observed relationship between brain structure, cognitive function, and IBS symptomatology suggests that effective interventions should target multiple domains simultaneously. Such a multifaceted approach recognizes IBS as a complex disorder requiring coordinated intervention across multiple domains.

Future research directions should expand upon these findings through multimodal investigation. The integration of functional neuroimaging, gut microbiome analysis, and broader clinical assessment [20] could provide a more comprehensive understanding of IBS pathophysiology. Longitudinal studies will be particularly crucial to determine the temporal relationship between brain changes and symptom development. Such studies would allow us to track the evolution of cognitive and structural alterations over time, identify early markers of disease progression, and evaluate the impact of various therapeutic interventions. This temporal perspective is essential for understanding whether observed brain changes represent the cause or consequence of IBS symptoms.

This comprehensive approach to understanding IBS aligns with the emerging paradigm of *precision medicine*. By considering the full spectrum of biological, cognitive, and behavioral manifestations, we may better identify patient subgroups and develop more personalized treatment strategies. The integration of brain structure, cognitive function, and clinical symptoms represents a promising framework for advancing both our understanding and treatment of this complex disorder. Ultimately, this integrated perspective may lead to more effective, personalized interventions that address the full range of IBS manifestations.

### 4.4. Limitations and Strengths: Critical Evaluation and Future Directions

Although this study contributes through its multimodal analytical approach, several of its limitations warrant discussion. The moderate sample size and lack of prospective data limit the generalizability, although our cohort is comparable to or larger than many neuroimaging studies in IBS. Moreover, we used cross-validation techniques and a holdout test data set as means to explore generalizability. The cross-sectional design, however, precludes inference about the causality or temporal dynamics of observed alterations. Additionally, while our machine learning approach achieved high sensitivity, the limited specificity suggests that brain structural and cognitive measures alone may be insufficient for definitive IBS diagnosis. Moreover, the moderate sample size particularly constrained our ability to conduct robust sex/gender-based analyses. This limitation is especially noteworthy given the evidence for substantial sex/gender differences in IBS presentation, progression, and treatment response [59]. The importance of sex/gender considerations in IBS research has become increasingly apparent. Clinical presentations show clear sex-based patterns, with IBS-C predominating in women and IBS-D in men [60]. These differences reflect complex interactions between biological and environmental factors. Sex hormones, particularly estrogen and progesterone, influence both gastrointestinal function and pain processing in the central nervous system [61]. Recent research has revealed sex-based differences extending to gut microbiota composition [62] and sensory processing. Notably, Labus et al. [21] demonstrated enhanced sensory sensitivity in women with IBS, potentially related to sex-specific morphometric variations in brain structure.

An inability to fully account for IBS symptom severity in our analyses was another limitation. Recent work by Li et al. [20] demonstrated that *symptom severity* correlates significantly with both cognitive performance and brain volumetric measures, particularly in regions associated with emotional processing and cognitive control. While chronic pain conditions can lead to progressive changes in pain-processing regions [26], establishing clear duration-related effects in IBS remains challenging due to symptom fluctuation and potential recall bias in *duration reporting*. These findings underscore the importance of incorporating both detailed symptom severity and duration measures in future studies to better characterize the relationship between clinical manifestations and brain–behavior patterns. We will also comment on choosing the threshold for the model’s prediction. Our initial approach used PyCaret’s default probability threshold of 0.5 for binary classification, where predictions ≥ 0.5 are classified as IBS (positive class) and <0.5 as HC (negative class). While our dataset has a class imbalance (63% IBS vs. 37% HC), which differs from epidemiological prevalence rates (∼10%), we maintained this default threshold to provide a baseline performance metric that is widely used and interpretable. This conservative choice of threshold (0.5 for a 63–37 split) likely means our reported performance metrics underestimate the model’s true discriminative ability. Future work could explore optimizing the threshold either to match our dataset’s class distribution or to align with epidemiological prevalence rates, potentially through methods such as ROC curve analysis, cost-sensitive learning approaches, and balancing techniques like Synthetic Minority Over-sampling TEchnique (SMOTE) (creating synthetic samples to balance classes) or class weights. This would be particularly relevant when adapting these models to populations with IBS prevalence closer to epidemiological rates.

Regarding brain morphometry, the collection of brain regions being studied was restricted to those reported by Skrobisz et al. [23] (cfr. Table A1). This is a limitation of the study, as several other brain regions have been shown to be involved in IBS, both by structural and functional MRI. For example, the insula and anterior cingulate cortex (ACC) play crucial roles in visceral sensation, pain processing, and emotional regulation in IBS [63]. The dorsolateral prefrontal cortex (dlPFC) is also implicated in cognitive flexibility and descending pain modulation in IBS [64]. Additionally, key nodes of the salience network, which are involved in detecting and filtering sensory information, may be affected in IBS [65]. Future research should investigate these and other brain regions to gain a more comprehensive understanding of brain morphometry in IBS.

The study presents several key methodological contributions to the field of irritable bowel syndrome research. Firstly, our methodological framework for systematic dual-version analysis on a fixed dataset provides a valuable template for future studies to assess the robustness of their findings across software versions. By analyzing our data this way, we can better distinguish between genuine biological differences and methodologically-induced variations, thereby strengthening the reliability of our findings about brain morphometric differences between HC and IBS groups. Related to the morphometric restrictions we made, our work also provides a new perspective and proof of concept toward “a next-generation histological atlas of the human brain for high-resolution neuroimaging studies” [66] in IBS. All regions mentioned above (e.g., insula, ACC, dlPFC) and their volumes, in addition to the subsegmentation of the hippocampus, thalamus, mesencephalon, pons, medulla oblongata, and more, are included in the segmentation results shown in Figure A2 (>300 regions in each hemisphere). This could be obtained for each subject in our cohort (albeit at a high computational cost) and used as morphometric features in prediction models.

A main contribution of this study is an advanced machine learning approach that integrates brain structural and cognitive measures, moving beyond traditional single-modal assessments. This type of computational methodology represents a significant advancement in the use of neuroimaging techniques in general, offering a more sophisticated analytical framework for understanding the complex neurological underpinnings of complex conditions such as IBS. By developing a machine learning model with high sensitivity, the study opens new avenues for more objective diagnostic strategies, even though the current specificity suggests the need for further refinement. Moreover, the study’s cohort and analytical approach contribute methodologically by establishing a robust dataset and openly available code that could be tested and further developed for future investigations. The computational neuroimaging methodology developed in this research, supporting data-driven approaches to understanding IBS from a neuro-cognitive perspective, has broader implications, potentially offering insights that could be applied to other neurological and psychiatric conditions with complex neuroimaging presentations.

## 5. Conclusions and Future Directions

The results point to several important directions for future research. First, larger-scale studies are needed to validate and extend our multivariate findings. Such studies should maintain rigorous methodological standards while increasing statistical power. Second, the standardization of neuroimaging analysis protocols, including the careful documentation of software versions and processing parameters, is crucial for reproducibility. Third, the field would benefit from the systematic investigation of how different analysis approaches might influence morphometric findings in IBS research. The observed version-dependent variations have critical implications for multi-site studies and meta-analyses. The preservation of relative group differences suggests that within-study comparisons remain valid, but that absolute measurements may not be directly comparable across studies using different FreeSurfer versions. This finding underscores the importance of harmonized processing pipelines in neuroimaging research, particularly for studies investigating subtle structural alterations in clinical populations. The observed systematic biases also highlight the need to carefully consider software version effects when conducting replication studies or meta-analyses of brain morphometry findings.

Overall, future studies should consider implementing standardized protocols for both imaging and cognitive assessment, facilitating meta-analyses and enabling more direct comparisons across studies. This standardization, combined with transparent reporting of methodological details, would strengthen the field’s ability to build cumulative knowledge about brain–gut interactions in IBS.

Longitudinal studies represent a particularly important future direction. Such studies could address crucial questions about the temporal dynamics of brain–gut interactions in IBS, including whether observed structural and cognitive changes precede or follow symptom development. Longitudinal data would also enable the better prediction of disease trajectories and treatment responses, potentially informing personalized interventions. These could include dietary modifications, e.g., a low-FODMAP diet, interventions tailored to individual microbiome profiles and trigger patterns, targeted cognitive interventions based on neuroplasticity patterns, or combined therapeutic approaches informed by temporal symptom patterns. The combination of longitudinal design with multimodal assessment would be particularly powerful, integrating structural and functional brain imaging, cognitive testing, gut microbiome profiling, immune biomarkers, metabolomics, and detailed symptom characterization. This comprehensive approach could not only provide unprecedented insights into IBS pathophysiology but also identify distinct patient subgroups with different underlying mechanisms, enabling more precise therapeutic targeting. Additionally, tracking the temporal relationships between central and peripheral alterations could reveal critical windows for therapeutic intervention and help establish causal relationships between observed changes, moving beyond the current correlational understanding of brain–gut interactions in IBS.

## Figures and Tables

**Figure 1 diagnostics-15-00470-f001:**
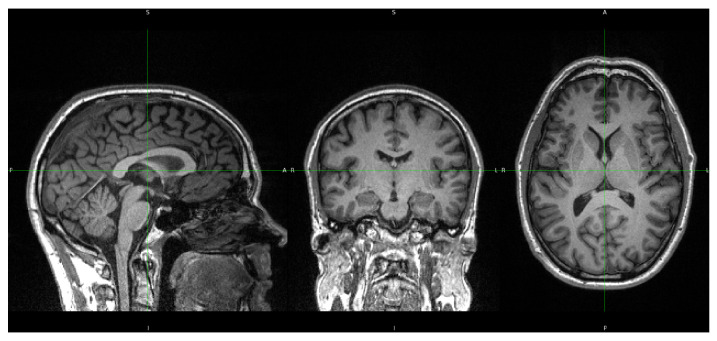
A 3D T1-weighted MPRAGE recording from BGA_046. The panels, from left to right, show the sagittal, coronal, and axial sections, respectively. Generated by: https://github.com/arvidl/ibs-brain/blob/main/notebooks/01-freesurfer-freeview-t1-aseg-bga-046.ipynb (accessed on 11 February 2025).

**Figure 2 diagnostics-15-00470-f002:**
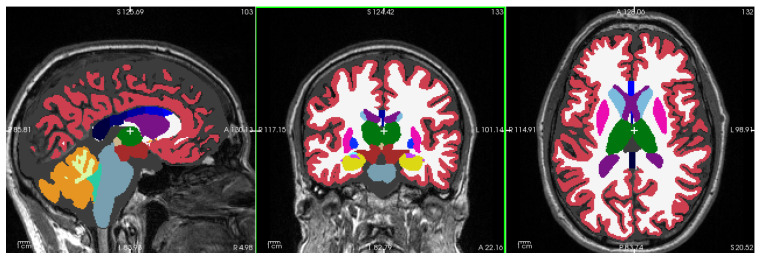
The color-coded aseg segmentation mask created by FreeSurfer 7.4.1 overlaid on 3D T1-w MPRAGE from BGA_046.The panels, from left to right, show the sagittal, coronal, and axial sections, respectively. The white cross is located in the medial part of left thalamus. Thalamus: green, hippocampus: yellow, caudate: light blue, putamen: pink, pallidum: purple, cortex: red, white matter: white. See also Figure A2. Generated by: https://github.com/arvidl/ibs-brain/blob/main/notebooks/01-freesurfer-freeview-t1-aseg-bga-046.ipynb (accessed on 11 February 2025).

**Figure 3 diagnostics-15-00470-f003:**
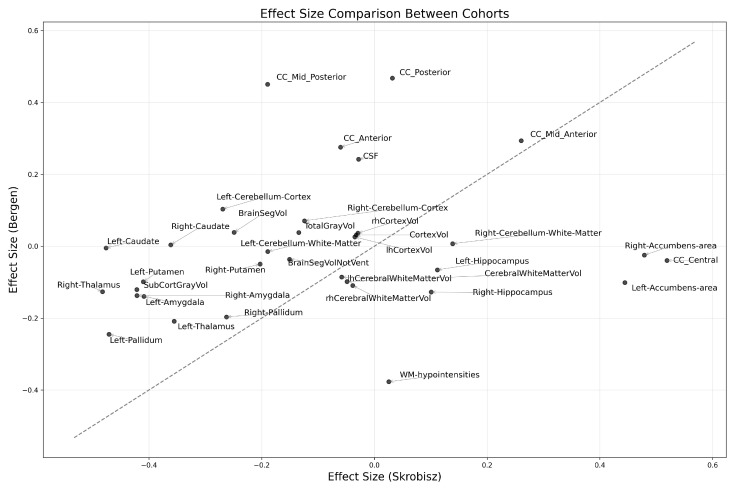
The large disparity in region-wise effect sizes for IBS versus HC, comparing the Skrobisz (2022) cohort and the Bergen cohort. Scatterplot of calculated Cohen’s d effect sizes for each region in both cohorts (see text for details). Generated by: https://github.com/arvidl/ibs-brain/blob/main/notebooks/03-replication-analysis-fs6.ipynb (accessed on 11 February 2025).

**Figure 4 diagnostics-15-00470-f004:**
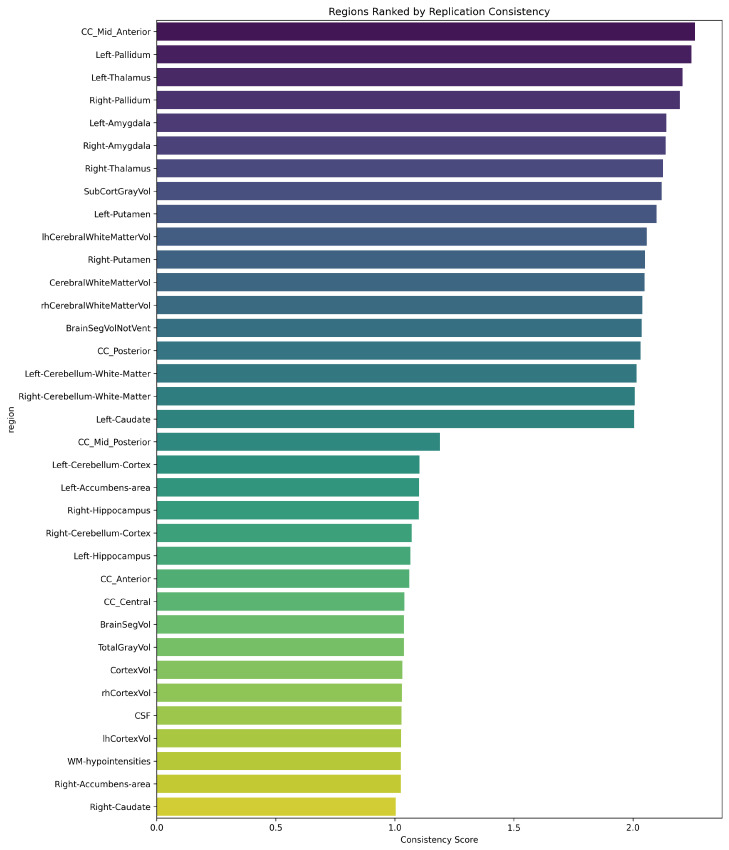
Brain regions ranked by cross-cohort reproducibility scores. The composite score (*S*) integrates directional consistency (σ), confidence interval overlap (ω), and effect magnitude (ϵ) between the Skrobisz and Bergen cohorts. Higher scores indicate the greater reproducibility of IBS-related volumetric alterations across independent samples. See the text for detailed scoring methodology. Generated by: https://github.com/arvidl/ibs-brain/blob/main/notebooks/03-replication-analysis-fs6.ipynb (accessed on 11 February 2025).

**Figure 5 diagnostics-15-00470-f005:**
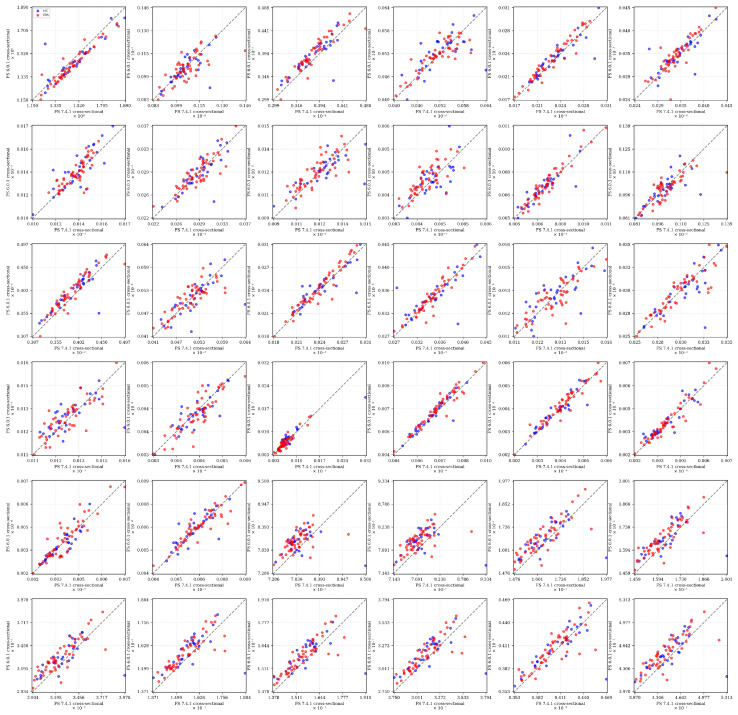
Comparison of FreeSurfer-derived regional brain volumes across versions 6.0.1 and 7.4.1 (cross-sectional processing). Scatter plots show eTIV-normalized volumes for each brain region, with version 6.0.1 values on the *y*-axis versus version 7.4.1 on the *x*-axis. The eTIV-volume [mm^3^] is shown in the upper left panel. Blue and red markers denote healthy controls and IBS patients, respectively. Identity lines indicate perfect cross-version agreement. See text for detailed analysis. Generated by: https://github.com/arvidl/ibs-brain/blob/main/notebooks/04-comparing-FS-versions-on-same-dataset.ipynb (accessed on 11 February 2025).

**Figure 6 diagnostics-15-00470-f006:**
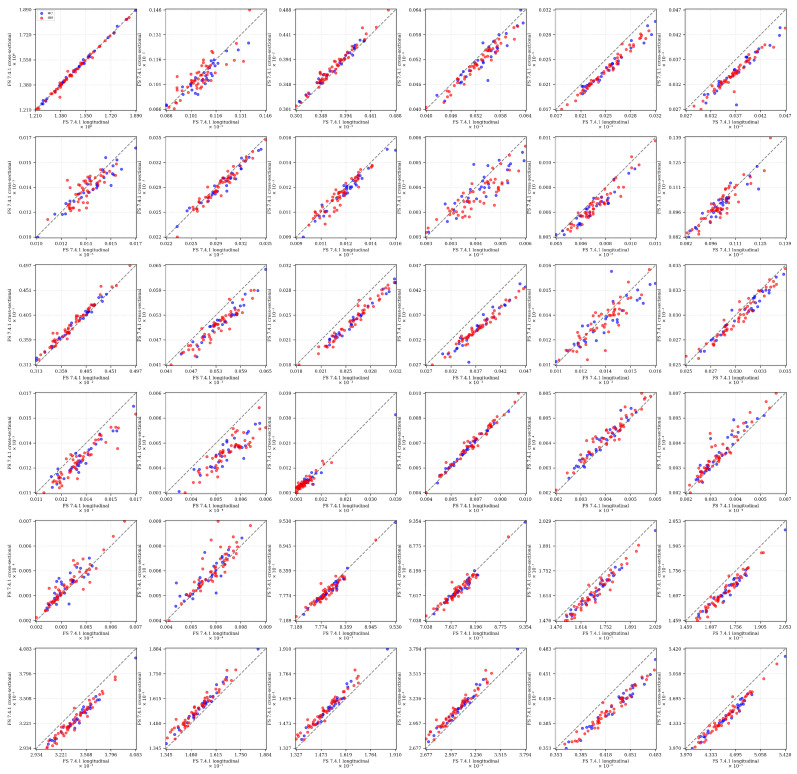
Comparison of regional brain volumes between FreeSurfer 7.4.1 cross-sectional and longitudinal processing streams. Scatter plots show eTIV-normalized volumes [eTIV in mm^3^] for each brain region, with cross-sectional values on the *y*-axis versus longitudinal values on the *x*-axis. Blue and red markers denote healthy controls and IBS patients, respectively. Identity lines indicate perfect cross-stream agreement. See text for detailed analysis. Generated by: https://github.com/arvidl/ibs-brain/blob/main/notebooks/04-comparing-FS-versions-on-same-dataset.ipynb (accessed on 11 February 2025).

**Figure 7 diagnostics-15-00470-f007:**
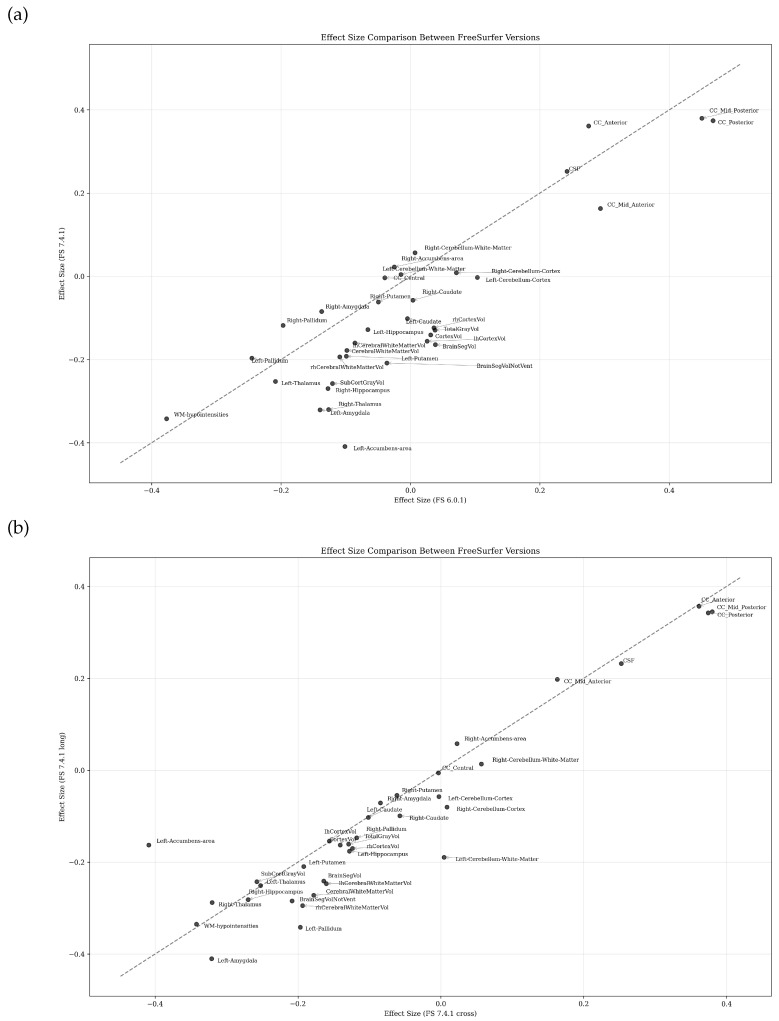
Effect sizes (Cohen’s d) of IBS versus healthy control group differences across brain regions: comparison of FreeSurfer methodological variants. (**a**) Cross-sectional processing stream comparison between FreeSurfer versions 6.0.1 and 7.4.1. (**b**) Processing stream comparison within FreeSurfer 7.4.1 (cross-sectional versus longitudinal). See text for detailed analysis. Generated by: https://github.com/arvidl/ibs-brain/blob/main/notebooks/04-comparing-FS-versions-on-same-dataset.ipynb (accessed on 11 February 2025).

**Figure 8 diagnostics-15-00470-f008:**
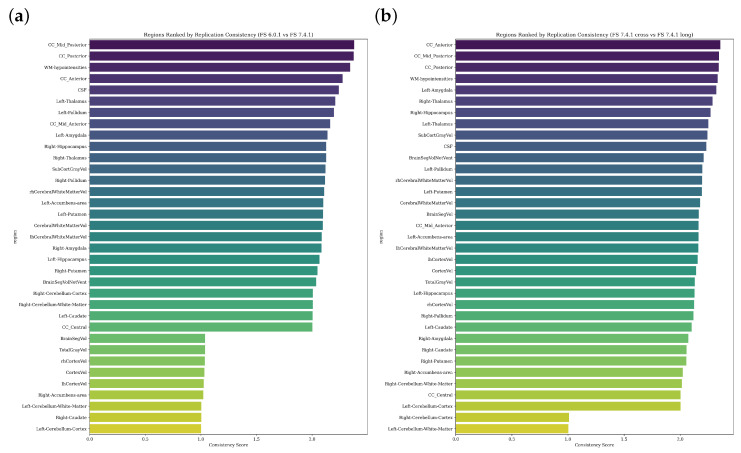
Brain regions ranked by reproducibility of IBS versus healthy control differences across FreeSurfer methodological variants. Composite scores (*S*) combine directional consistency (σ), confidence interval overlap (ω), and effect magnitude (ϵ). Panel (**a**) compares FreeSurfer versions 6.0.1 and 7.4.1 (cross-sectional); panel (**b**) contrasts cross-sectional and longitudinal processing streams within FreeSurfer 7.4.1. See text for detailed analysis. Generated by: https://github.com/arvidl/ibs-brain/blob/main/notebooks/04-comparing-FS-versions-on-same-dataset.ipynb (accessed on 11 February 2025).

**Figure 9 diagnostics-15-00470-f009:**
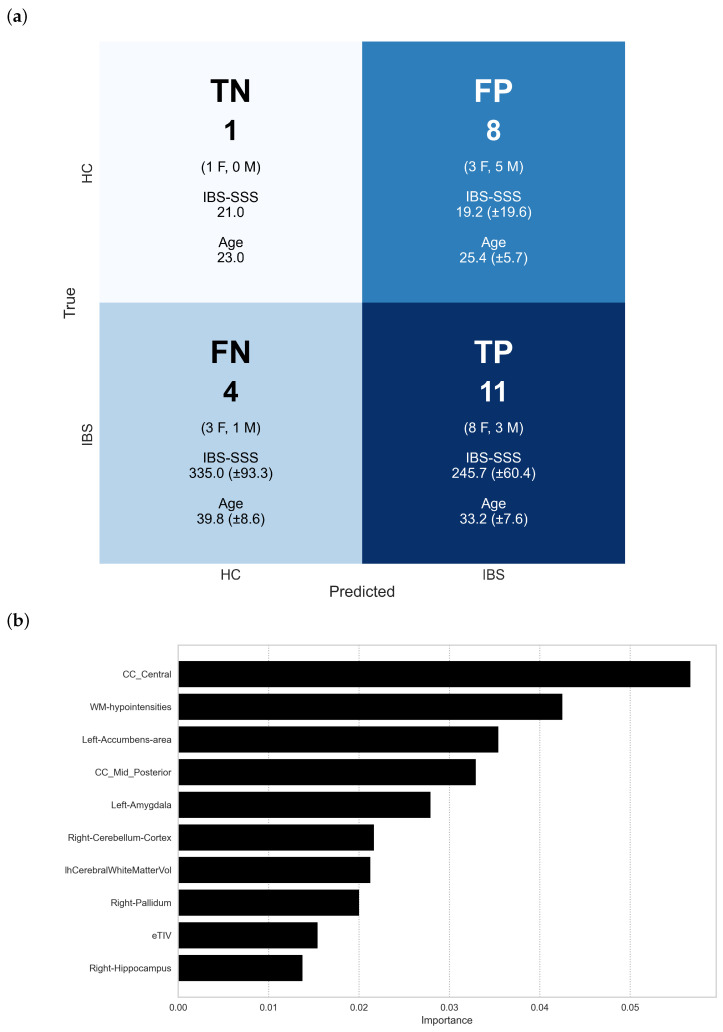
Machine learning-based discrimination between IBS and healthy controls using brain morphometry. (**a**) Confusion matrix showing prediction outcomes from XGBoost classification on the test dataset, with quadrants indicating true negatives (TN), false positives (FP), false negatives (FN), and true positives (TP). (**b**) Ten most discriminative brain regions identified through permutation importance analysis in the XGBoost model. Generated by: https://github.com/arvidl/ibs-brain/blob/main/notebooks/05-predicting-IBS-vs-HC-from-morphometric-measures.ipynb (accessed on 11 February 2025).

**Figure 10 diagnostics-15-00470-f010:**
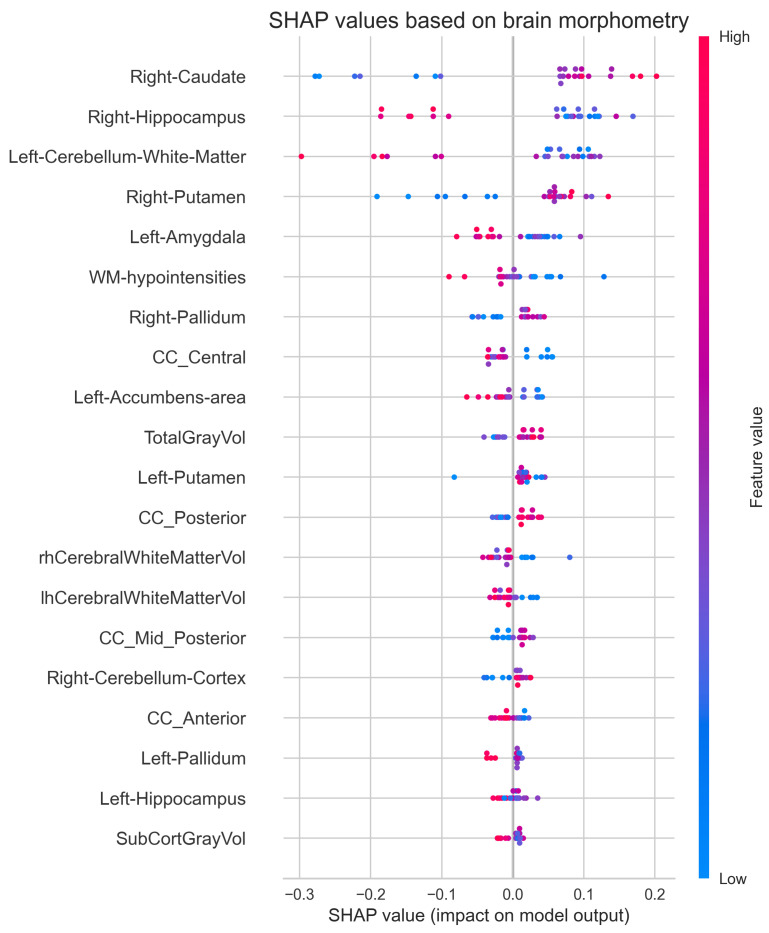
Feature contribution analysis using SHAP values for brain morphometry-based classification. SHAP values (*x*-axis) quantify each region’s impact on classification probability, with negative values favoring healthy control classification and positive values favoring IBS. Each point represents a single prediction, with its color indicating the relative magnitude of the morphometric measurement (red: high values, blue: low values). The distribution of points reveals how feature values influence model predictions across individual cases. Generated by: https://github.com/arvidl/ibs-brain/blob/main/notebooks/05-predicting-IBS-vs-HC-from-morphometric-measures.ipynb (accessed on 11 February 2025).

**Figure 11 diagnostics-15-00470-f011:**
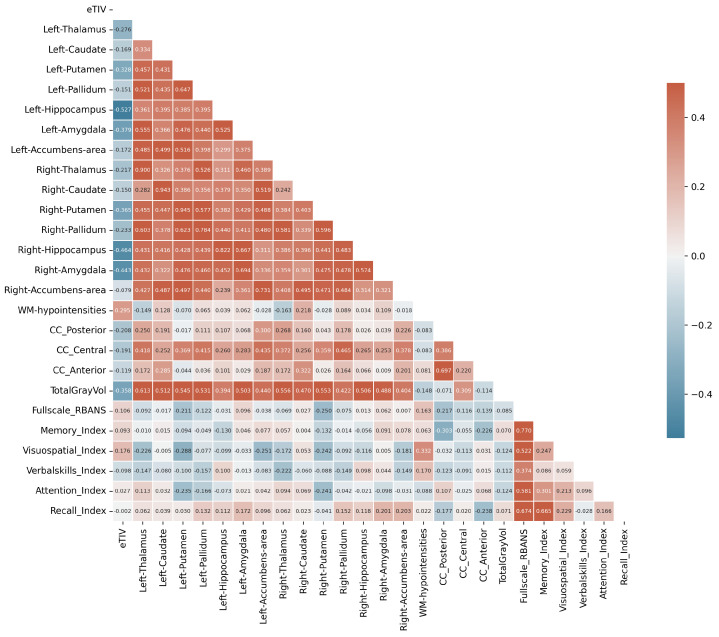
Brain structure–cognition relationships visualized through Spearman rank correlations. The correlation matrix shows pairwise associations between regional brain volumes (eTIV-normalized) and RBANS cognitive scores. Red indicates positive correlations, blue indicates negative correlations, with color intensity reflecting correlation strength. Generated by: https://github.com/arvidl/ibs-brain/blob/main/notebooks/06-morphometry-cognition-exploration.ipynb (accessed on 11 February 2025).

**Figure 12 diagnostics-15-00470-f012:**
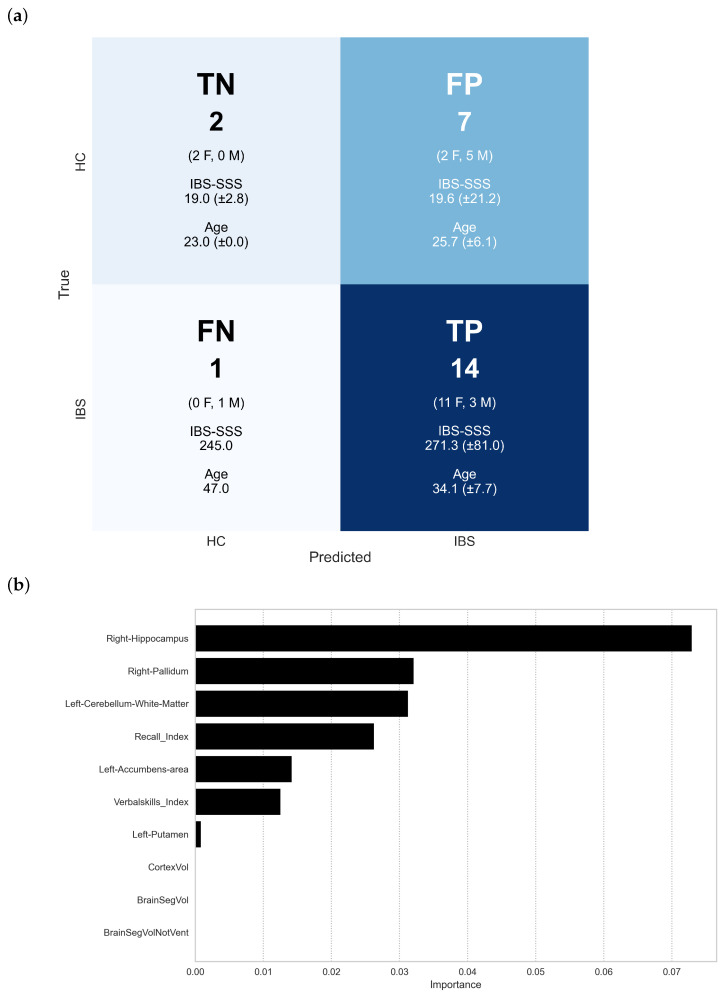
Multimodal machine learning classification of IBS combining brain morphometry and cognitive measures. (**a**) Confusion matrix from XGBoost model testing, showing classification outcomes with detailed participant characteristics per quadrant (TN: true negatives, FP: false positives, FN: false negatives, TP: true positives). Gender distribution (F: female, M: male), IBS-SSS scores, and age are reported for each category. (**b**) Feature importance ranking derived from the model, showing relative contribution of brain structural and cognitive measures to classification decisions. Generated by: https://github.com/arvidl/ibs-brain/blob/main/notebooks/07-predicting-IBS-vs-HC-from-morphometry-and-cognition.ipynb (accessed on 11 February 2025).

**Figure 13 diagnostics-15-00470-f013:**
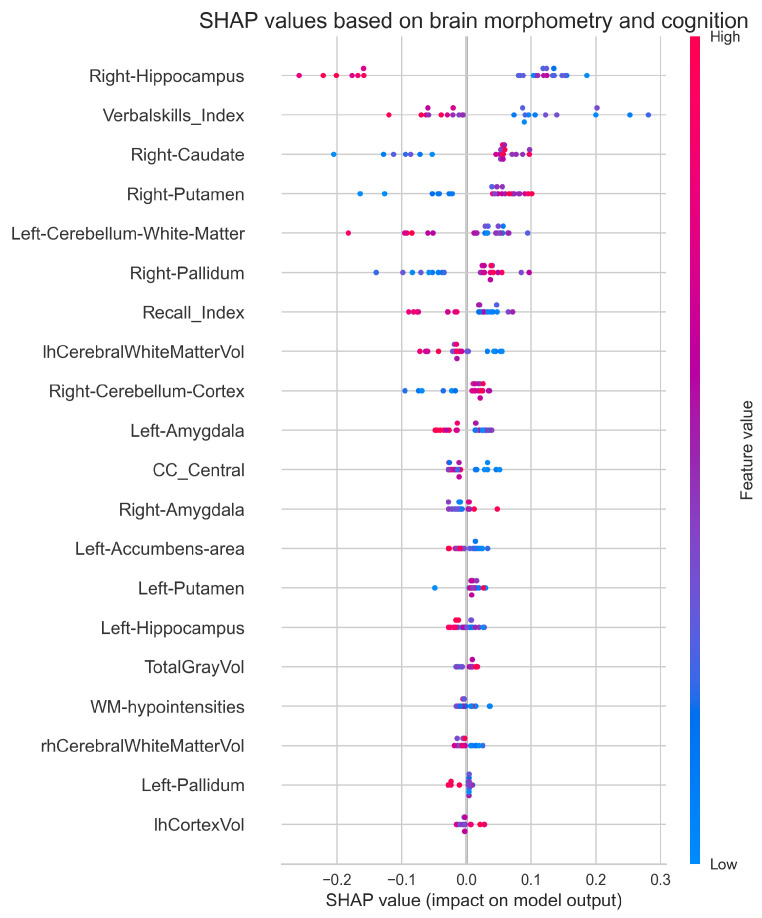
Feature contribution analysis using SHAP values for multimodal brain morphometry and cognition-based classification. SHAP values (*x*-axis) quantify each region’s impact on classification probability, with negative values favoring healthy control classification and positive values favoring IBS. Each point represents a single prediction, with color indicating the relative magnitude of the morphometric measurement (red: high values, blue: low values). The distribution of points reveals how feature values influence model predictions across individual cases. Generated by: https://github.com/arvidl/ibs-brain/blob/main/notebooks/07-predicting-IBS-vs-HC-from-morphometry-and-cognition.ipynb (accessed on 11 February 2025).

**Table 1 diagnostics-15-00470-t001:** Exclusion and inclusion criteria for the IBS patients. Source: Retrieved from [24].

Inclusion Criteria	Exclusion Criteria
Rome-IV criteria: Recurrent abdominal pain on average at least 1 day/week during the last 3 months and associated alterations in bowel habits at least 6 months before diagnosis. Other causes are excluded.	Pharmacological treatment affecting GI tract, including medication for anxiety and depression, diabetes, coeliac disease, Inflammatory bowel disease, polycystic ovary syndrome, active Helicobacter pylori infection, Parkinson’s disease, amyotrophic lateral sclerosis, or psychiatric disorders.
Normal diet for at least 3 weeks before inclusion	
IBS score equal to or above 175	Treated with antibiotics for the last 3 months
	Diets such as vegetarian or vegan
	Use of probiotics or low-FODMAP diet within the last 3 weeks
	Previous intestinal surgery, except appendectomy
	Metallic implants, claustrophobia, incompatible with MRI
	Travel outside Europe in the last 3 weeks
	Plans to travel in the near future
	Pregnancy

**Table 2 diagnostics-15-00470-t002:** Demographic and clinical characteristics of the study sample. Age is reported in years; IQR = interquartile range; F/M = female/male ratio expressed as percentages. Generated by: https://github.com/arvidl/ibs-brain/blob/main/notebooks/02-demographics-and-clinical-characteristics.ipynb (accessed on 11 February 2025).

Group	Age	IBS-SSS	Sex	N	Missing
	Median (IQR)	Median (IQR)	F/M (%)		IBS-SSS
HC (N = 29)	33.0 (23.0)	21.0 (30.0)	69.0/31.0	29	3
IBS (N = 49)	34.0 (14.0)	264.0 (95.0)	77.6/22.4	49	3

**Table 3 diagnostics-15-00470-t003:** Comparison of eTIV-normalized regional brain volumes between the two cohorts.

	Skrobisz Cohort (FS 6.0)				Bergen Cohort (FS 6.0.1)			
Brain Region	HC (N = 19)		IBS (N = 20)		HC (N = 29)		IBS (N = 49)	
	Mean	SD	Mean	SD	Mean	SD	Mean	SD
Left Cerebellum WM	0.00992	0.00113	0.00971	0.00107	0.01050	0.00092	0.01048	0.00092
Left Cerebellum Cortex	0.03628	0.00302	0.03553	0.00256	0.03894	0.00344	0.03931	0.00373
Left Thalamus	0.00511	0.00037	0.00500	0.00024	0.00523	0.00046	0.00514	0.00039
Left Caudate	0.00239	0.00025	0.00228	0.00021	0.00236	0.00026	0.00236	0.00031
Left Putamen	0.00336	0.00033	0.00324	0.00028	0.00348	0.00038	0.00344	0.00039
Left Pallidum	0.00140	0.00012	0.00135	0.00010	0.00140	0.00015	0.00137	0.00011
Left Hippocampus	0.00270	0.00021	0.00272	0.00020	0.00291	0.00027	0.00290	0.00024
Left Amygdala	0.00118	0.00013	0.00113	0.00015	0.00122	0.00010	0.00120	0.00010
Left Accumbens Area	0.00031	0.00005	0.00034	0.00006	0.00043	0.00007	0.00042	0.00006
CSF	0.00061	0.00009	0.00060	0.00012	0.00067	0.00012	0.00070	0.00014
Right Cerebellum WM	0.00908	0.00106	0.00922	0.00100	0.00997	0.00089	0.00998	0.00085
Right Cerebellum Cortex	0.03652	0.00321	0.03616	0.00264	0.03972	0.00344	0.03998	0.00376
Right Thalamus	0.00488	0.00030	0.00475	0.00024	0.00512	0.00044	0.00507	0.00036
Right Caudate	0.00244	0.00024	0.00236	0.00024	0.00244	0.00024	0.00244	0.00030
Right Putamen	0.00336	0.00030	0.00330	0.00028	0.00351	0.00037	0.00349	0.00035
Right Pallidum	0.00136	0.00012	0.00133	0.00010	0.00132	0.00013	0.00130	0.00011
Right Hippocampus	0.00282	0.00022	0.00285	0.00021	0.00301	0.00024	0.00298	0.00023
Right Amygdala	0.00125	0.00012	0.00120	0.00012	0.00128	0.00009	0.00127	0.00010
Right Accumbens Area	0.00034	0.00004	0.00036	0.00005	0.00043	0.00005	0.00043	0.00006
WM Hypointensities	0.00047	0.00015	0.00048	0.00013	0.00079	0.00031	0.00069	0.00025
CC Posterior	0.00065	0.00013	0.00065	0.00010	0.00065	0.00010	0.00070	0.00011
CC Mid Posterior	0.00038	0.00007	0.00036	0.00007	0.00037	0.00007	0.00040	0.00007
CC Central	0.00039	0.00009	0.00043	0.00008	0.00039	0.00009	0.00039	0.00010
CC Mid Anterior	0.00041	0.00009	0.00044	0.00013	0.00038	0.00008	0.00041	0.00011
CC Anterior	0.00062	0.00010	0.00061	0.00008	0.00062	0.00010	0.00065	0.00010
BrainSegVol	0.75340	0.01784	0.74913	0.01647	0.80464	0.02487	0.80558	0.02397
BrainSegVolNotVent	0.74137	0.01880	0.73857	0.01836	0.79224	0.02511	0.79132	0.02490
lhCortexVol	0.15339	0.00620	0.15313	0.00871	0.16670	0.00800	0.16693	0.00951
rhCortexVol	0.15490	0.00690	0.15467	0.00859	0.16614	0.00828	0.16646	0.00939
CortexVol	0.30829	0.01298	0.30780	0.01715	0.33283	0.01611	0.33339	0.01880
lhCerebralWhiteMatterVol	0.15101	0.00748	0.15058	0.00742	0.15990	0.00858	0.15915	0.00876
rhCerebralWhiteMatterVol	0.15103	0.00757	0.15075	0.00727	0.15925	0.00829	0.15827	0.00938
CerebralWhiteMatterVol	0.30205	0.01500	0.30133	0.01461	0.31915	0.01678	0.31742	0.01808
SubCortGrayVol	0.03930	0.00194	0.03855	0.00162	0.04092	0.00258	0.04063	0.00236
TotalGrayVol	0.42105	0.01376	0.41884	0.01868	0.45307	0.02208	0.45396	0.02432

Note: All volumes are normalized to estimated total intracranial volume (eTIV). HC = healthy controls; IBS = irritable bowel syndrome; SD = standard deviation; WM = white matter. Generated by: https://github.com/arvidl/ibs-brain/blob/main/notebooks/03-replication-analysis-fs6.ipynb (accessed on 11 February 2025).

**Table 4 diagnostics-15-00470-t004:** A non-parametric analysis comparing cognitive features in the IBS and HC groups. Values presented as median (interquartile range, IQR). Group differences assessed using Mann–Whitney U tests (uncorrected *p*-values), where we multiply by 6 to get the corrected values. Effect sizes quantified using Cliff’s delta, where positive values indicate higher scores in healthy controls. Generated by: https://github.com/arvidl/ibs-brain/blob/main/notebooks/06-morphometry-cognition-exploration.ipynb (accessed on 11 February 2025).

Variable	HC	IBS	*p*-Value	Cliff’s Delta
Full-scale RBANS	103.0 (93.0–108.0)	91.0 (85.0–100.0)	0.002	0.213
Memory Index	100.0 (86.0–109.0)	86.0 (81.0–105.0)	0.031	0.147
Visuospatial Index	97.0 (90.0–107.0)	96.0 (90.0–105.0)	0.763	0.021
Verbal Skills Index	105.0 (95.0–113.0)	95.0 (89.0–111.0)	0.087	0.116
Attention Index	98.0 (89.0–108.0)	97.0 (83.0–101.0)	0.118	0.107
Recall Index	107.0 (92.0–113.0)	95.0 (85.0–100.0)	0.006	0.186

**Table 5 diagnostics-15-00470-t005:** Classification performance metrics comparing XGBoost models trained on morphometric features alone (M) versus combined morphometric and cognitive features (M ∪ C). Metrics include sensitivity (TPR), specificity (TNR), precision (PPV), accuracy (ACC), and additional measures of classification reliability. M: morphometric; C: cognitive features. See list of abbreviations for the rest of column names denoting 11 different metrics. Generated by: https://github.com/arvidl/ibs-brain/blob/main/notebooks/07-predicting-IBS-vs-HC-from-morphometry-and-cognition.ipynb (accessed on 11 February 2025).

Feature Set	TPR	TNR	PPV	NPV	FPR	FNR	FDR	ACC	BACC	F1	MCC
M	0.733	0.111	0.579	0.200	0.889	0.267	0.421	0.500	0.422	0.647	−0.185
M ∪ C	0.933	0.222	0.667	0.667	0.778	0.067	0.333	0.667	0.578	0.778	0.228

## Data Availability

The complete analysis workflow is publicly available at https://github.com/arvidl/ibs-brain (accessed on 11 February 2025) comprising reproducible Jupyter notebooks containing all analysis code and visualizations, cleaned datasets in .CSV format, Conda environment configuration for exact replication, and source code for generating all tables and figures presented in the Results section. The computational analyses were developed with assistance from the Claude 3.5 Sonnet (Anthropic, San Francisco, CA, USA) large language model integrated within the Cursor (Anysphere Inc., San Francisco, California, USA) AI code editor and development environment.

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
