# Peer review of "Brain Morphometry and Cognitive Features in the Prediction of Irritable Bowel Syndrome"

_diagnostics, 2025, doi:10.3390/diagnostics15040470_

Round 1
Reviewer 1 Report
Comments and Suggestions for Authors
This study aims to replicate previous morphometric findings in irritable bowel syndrome (IBS) patients, compare software, and investigate if multivariate analysis of brain morphometry and cognitive performance can distinguish IBS patients from healthy controls (HCs). A dataset composed of data from 49 IBS patients and 29 HCs using structural brain MRI is used in the numerical experiments. Combining brain morphometry and cognitive measures improves IBS-HC discrimination. The importance of subcortical structures and specific cognitive domains supports complex brain-gut interactions in IBS, highlighting the need for multimodal approaches and rigorous methodologies.
In general, the paper is very well written, discussing non-trivial methods for a practically relevant problem. The contributions in the paper are clear, and the literature overview is sound. However, I have a few important remarks:
- Line 208 – the correct link for the GitHub repo is actually https://github.com/arvidl/ibs-brain (and not https://arvidl.github.com/ibs-brain).
- Table from Figure A1: I recommend including the confidence intervals associated with the average values that the authors have reported (in order to support a statistical comparison of the classifiers’ performances).
- If I understood correctly, the numerical experiments would correspond to a retrospective study. It would be beneficial to have some results for a prospective study, where the best classifier is applied to new data and the predicted classes/label are validated by the medical specialists.
Author Response
This study aims to replicate previous morphometric findings in irritable bowel syndrome (IBS) patients, compare software, and investigate if multivariate analysis of brain morphometry and cognitive performance can distinguish IBS patients from healthy controls (HCs). A dataset composed of data from 49 IBS patients and 29 HCs using structural brain MRI is used in the numerical experiments. Combining brain morphometry and cognitive measures improves IBS-HC discrimination. The importance of subcortical structures and specific cognitive domains supports complex brain-gut interactions in IBS, highlighting the need for multimodal approaches and rigorous methodologies.
In general, the paper is very well written, discussing non-trivial methods for a practically relevant problem. The contributions in the paper are clear, and the literature overview is sound.
Answer: Thank you.
However, I have a few important remarks:
- Line 208 – the correct link for the GitHub repo is actually https://github.com/arvidl/ibs-brain (and not https://arvidl.github.com/ibs-brain).
- Table from Figure A1: I recommend including the confidence intervals associated with the average values that the authors have reported (in order to support a statistical comparison of the classifiers’ performances).
- If I understood correctly, the numerical experiments would correspond to a retrospective study. It would be beneficial to have some results for a prospective study, where the best classifier is applied to new data and the predicted classes/label are validated by the medical specialists
Answer:
1. Thank you for informing us of the error in the link to the GitHub repo. It is corrected in the revised manuscript on page 6: https://github.com/arvidl/ibs-brain.
2. We added means and std of selected classifiers and metrics to the Figure A1 legend. Further interpretation and clarification was provided in the following text (page 49) :
In our comparative analysis of multiple models (XGBoost, Dummy, LDA, ADA, and RF), we employed bootstrap resampling to evaluate three metrics: Accuracy, AUC, and F1-score. After applying Bonferroni correction for multiple comparisons, we found no statistically significant differences between the models.
For the IBS versus HC classification task, the dummy classifier established a baseline accuracy by predicting based on class distribution (pbaseline = nIBS / ntotal ~ 0.6282). Any sophisticated classifier should exceed this baseline to demonstrate meaningful predictive capability. The underperformance of LDA relative to the baseline suggests potential violations of its core assumptions, specifically the normal distribution of features and equal covariance matrices between IBS and HC groups. Alternatively, the relationship between morphometric measures and IBS status may be more complex than linear discrimination can capture. The Ada Boost (ADA) classifier's suboptimal performance, despite its sophisticated ensemble learning approach, may be attributed to overfitting to noise in our limited dataset (n = 78) of morphometric measures, with the boosting process potentially amplifying misclassified outliers. Additionally, the relationship between brain structure and IBS status may be too subtle or complex for the boosting algorithm to capture effectively. To address these limitations, we expanded our feature set by incorporating cognitive performance metrics alongside the morphometric predictors).
3 We fully agree that prospective data would strongly improve the study. Unfortunately, such data were not available to us. We have, therefore, included machine learning procedures to enhance the generalization of results in our cross-sectional study. A comment on this limitation is included in the section “Limitations and strengths: Critical evaluation and future directions” (page 35):
The moderate sample size and lack of prospective data will limit generalizability, although our cohort is comparable to or larger than many neuroimaging studies in IBS. Moreover, we used cross-validation techniques and a holdout test data set as means to explore generalizability.

Reviewer 2 Report
Comments and Suggestions for Authors
1) “We included only participants with complete key measures and high-quality MRI scans suitable for automated brain segmentation, optimizing data quality while maximizing sample size” I have a concern that there could be some data bias since you intentionally selected a subset of all available participants.
2) “Almost all HCs obtained an IBS-SSS score at the lowest level (< 75), with some reporting a score within the mild level ([75, 175))” Please use exact numbers rather than vague descriptions.
3) “it allows us to assess the impact of software evolution on morphometric measurements on a fixed dataset” I’m not sure if the evaluation on the evolution of software is important or not, since it is not the author’s contribution. In addition, it should be a common sense that the newer software should be better than the elder version, and thus the evaluation seems not that valuable.
4) I saw you have accuracy measurement, how did you choose the threshold for each model?
5) From Figure A1, I believe the authors didn’t train ML models in a correct way, or, didn’t tune the models for the best possible performance. The reasons are that I have observed that the lda method has AUC=0.5, which equals to random guess, and the ada method has AUC significantly < 0.5, which even worse than random guess.
Author Response
1) “We included only participants with complete key measures and high-quality MRI scans suitable for automated brain segmentation, optimizing data quality while maximizing sample size” I have a concern that there could be some data bias since you intentionally selected a subset of all available participants.
Answer: Thank you for raising this important methodological point. To be fully transparent about our participant selection process in relation to the Bergen Brain-Gut project, we have added the following (page 3):
The Bergen Brain-Gut project's initial cohort consisted of 85 subjects with baseline MRI scans. Our final analytical sample of 78 participants (92\% inclusion rate) was determined by predefined criteria. The seven excluded participants consisted of four subjects lacking RBANS test results, one participant was excluded due to non-Norwegian language proficiency affecting cognitive testing validity, and two subjects had incomplete datasets (one IBS patient, one healthy control). These exclusions were based on missing data or predefined quality criteria rather than post-hoc selection, and the balanced distribution across patient and control groups suggests minimal risk of systematic bias.
This information is included when describing the patients on page 3.
2) “Almost all HCs obtained an IBS-SSS score at the lowest level (< 75), with some reporting a score within the mild level ([75, 175))” Please use exact numbers rather than vague descriptions.
Answer: We added (page 4 in the revised manuscript):
Of the 29 HC participants, 26 (89.7%) obtained an IBS-SSS score below 75 (lowest level), while 0 (0.0%) reported scores between 75 and 175 (mild level). The median IBS-SSS score for the HC group was 21.0 (IQR: 9.8-39.8), with a maximum score of 69.0 in this group.
3) “it allows us to assess the impact of software evolution on morphometric measurements on a fixed dataset” I’m not sure if the evaluation on the evolution of software is important or not, since it is not the author’s contribution. In addition, it should be common sense that the newer software should be better than the elder version, and thus, the evaluation seems not that valuable.
Answer: Although we agree that newer software should be better than the older version, we still find it important to directly use the dual-version analysis as a way to strengthen our findings rather than as a software evaluation exercise. For both versions, we focused on the automated segmentation of subcortical structures using FreeSurfer's aseg pipeline, which identifies and quantifies the volume of distinct brain regions (detailed in Table A1). This dual-version approach serves multiple critical purposes: first, it enables direct comparison with Skrobisz et al.'s (2022) findings, ensuring reproducibility and comparability with existing literature. Second, while newer software versions generally offer improvements, studies have shown that version changes can introduce systematic differences in morphometric measurements (Klauschen 2009, Jovicich 2009, Gronenschild 2012, Glatard 2015, Knussmann 2022, Debiasi 2023), which could potentially affect the interpretation of disease-related differences.
We added the following to the discussion part (page 36):
Firstly, our methodological framework for systematic dual-version analysis on a fixed dataset provides a valuable template for future studies to assess the robustness of their findings across software versions. By analyzing our data this way, we can better distinguish between genuine biological differences and methodologically-induced variations, thereby strengthening the reliability of our findings about brain morphometric differences between HC and IBS groups.
4) I saw you have accuracy measurement, how did you choose the threshold for each model?
Answer: We appreciate this comment and have added the following to the Discussion (page 36):
We will also comment on choosing the threshold for the model’s prediction. Our initial approach used PyCaret's default probability threshold of 0.5 for binary classification, where predictions greater than or equal to 0.5 are classified as IBS (positive class) and < 0.5 as HC (negative class). While our dataset has a class imbalance (63% IBS vs. 37% HC), which differs from epidemiological prevalence rates (~ 10%), we maintained this default threshold to provide a baseline performance metric that is widely used and interpretable. This conservative choice of threshold (0.5 for a 63-37 split) likely means our reported performance metrics are underestimating the model's true discriminative ability. Future work could explore optimizing the threshold either to match our dataset's class distribution or to align with epidemiological prevalence rates, potentially through methods such as ROC curve analysis, cost-sensitive learning approaches, and balancing techniques like SMOTE (create synthetic samples to balance classes) or class weights. This would be particularly relevant when adapting these models to populations with IBS prevalence closer to epidemiological rates.
5) From Figure A1, I believe the authors didn’t train ML models in a correct way, or, didn’t tune the models for the best possible performance. The reasons are that I have observed that the lda method has AUC=0.5, which equals to random guess, and the ada method has AUC significantly < 0.5, which even worse than random.
Answer: We added means and std of selected classifiers and metrics to the Figure A1 legend. Further interpretation and clarification was provided in the following text (page 49-50) :
In our comparative analysis of multiple models (XGBoost, Dummy, LDA, ADA, and RF), we employed bootstrap resampling to evaluate three metrics: Accuracy, AUC, and F1-score. After applying Bonferroni correction for multiple comparisons, we found no statistically significant differences between the models.
For the IBS versus HC classification task, the dummy classifier established a baseline accuracy by predicting based on class distribution (pbaseline = nIBS / ntotal ~ 0.6282). Any sophisticated classifier should exceed this baseline to demonstrate meaningful predictive capability. The underperformance of LDA relative to the baseline suggests potential violations of its core assumptions, specifically the normal distribution of features and equal covariance matrices between IBS and HC groups. Alternatively, the relationship between morphometric measures and IBS status may be more complex than linear discrimination can capture. The Ada Boost (ADA) classifier's suboptimal performance, despite its sophisticated ensemble learning approach, may be attributed to overfitting to noise in our limited dataset (n = 78) of morphometric measures, with the boosting process potentially amplifying misclassified outliers. Additionally, the relationship between brain structure and IBS status may be too subtle or complex for the boosting algorithm to capture effectively. To address these limitations, we expanded our feature set by incorporating cognitive performance metrics alongside the morphometric predictors.

Round 2
Reviewer 2 Report
Comments and Suggestions for Authors
Thanks the authors for the detail reply. The authors resolved some of my concerns, however, the 5th question was not resolved well. I highly doubt there could be a methodological incorrectness there, as the AUC of 'ada', 'ida' and 'ridge' method is <0.5. For example, 'ridge classifier' is basically a logistic regression with normalization term, which is very hard to achieve AUC of 0.2917.
Author Response
COMMENT: Thanks the authors for the detail reply.The authors resolved some of my concerns, however, the 5th question was not resolved well.
I highly doubt there could be a methodological incorrectness there, as the AUC of 'ada', 'ida'
and 'ridge' method is <0.5. For example, 'ridge classifier' is basically a logistic regression
with normalization term, which is very hard to achieve AUC of 0.2917.
RESPONSE:
We appreciate this important concern about the unexpectedly low AUC values (Figure A1). The observed AUC values below 0.5 for Ada Boost (0.3708), LDA (0.3792), and Ridge classifier (0.2917) indicate that these models learned legitimate patterns but systematically predicted the opposite class. This is supported by accuracies near 0.5 (ridge: 0.6067, lda: 0.5000, ada: 0.4867) and consistently negative or near-zero Kappa and MCC scores. Several factors contributed: Class imbalance in cross-validation folds led to systematic misclassification; in Ridge classification, the hyperparameter settings, in combination with regularization, may have led to a misalignment between the learned weights and the underlying data distribution, thereby contributing to the inversion; and AdaBoost, in small datasets, could have concentrated on outliers and amplified noise rather than true patterns. The small sample size and limited discriminative power of morphometric features further complicated model learning. We addressed these limitations by incorporating additional clinical variables (RBANS), as documented in Section "Multimodal Classification of IBS Using Brain Structure and Cognitive Measures" and in our notebook "07-predicting-IBS-vs-HC-from-morphometry-and-cognition.ipynb".
Round 3
Reviewer 2 Report
Comments and Suggestions for Authors
Thanks the authors for the prompt response. What I meant was, it seems like you didn't do any attempts to tune the hyperparameters of any of the machine learning baseline models you compared with, but just using the default hyperparameter settings from the code packages. This cannot justify why xgboost is the best among all candidate ML models, since you haven't tried to figure out what is the 'best possible performance' for other ML baseline models. This raised my concern about your experimental design from the ML side. For example, in Ridge Classifier, if you set the hyperparameter associated with L2 norm to be zero, then it is identical to a logistic regression (lr as you mentioned). This means that, if you get a 0.55 AUC by using logistic regression, then you will at least get a 0.55 if you use Ridge Classifier. This is because one of the possible hyperparameter setting is: lambda=0 (lambda is the hyperparameter controlling the ratio of the main objective function and the regularization term). If you have tried different settings about the lambda for Ridge classifier, then you must have tried lambda=0. If lambda=0, then the Ridge classifier becomes Logistic Regression. Therefore, the Ridge classifier should at least get an AUC of 0.55, but not 0.29.
Author Response
COMMENT:
... What I meant was, it seems like you didn't do any attempts to tune the hyperparameters of any of the machine learning baseline models you compared with, but just using the default hyperparameter settings from the code packages. This cannot justify why xgboost is the best among all candidate ML models, since you haven't tried to figure out what is the 'best possible performance' for other ML baseline models. This raised my concern about your experimental design from the ML side. For example, in Ridge Classifier, if you set the hyperparameter associated with L2 norm to be zero, then it is identical to a logistic regression (lr as you mentioned). This means that, if you get a 0.55 AUC by using logistic regression, then you will at least get a 0.55 if you use Ridge Classifier. This is because one of the possible hyperparameter setting is: lambda=0 (lambda is the hyperparameter controlling the ratio of the main objective function and the regularization term). If you have tried different settings about the lambda for Ridge classifier, then you must have tried lambda=0. If lambda=0, then the Ridge classifier becomes Logistic Regression. Therefore, the Ridge classifier should at least get an AUC of 0.55, but not 0.29.
RESPONSE:
Thank you for your insightful comment about hyperparameter tuning. We appreciate your attention to the Ridge Classifier's performance and have addressed this through additional analysis. We conducted comprehensive hyperparameter optimization, including testing α = 0 (equivalent to Logistic Regression) and various other α values, along with class weight balancing to address the IBS/HC imbalance. Our systematic grid search included solver variations ('auto', 'svd', 'cholesky') and class weights. The optimal parameters (α = 0, class_weight = {0: 1.35, 1: 0.794}) yielded a mean AUC of 0.529 (±0.268), demonstrating that even with extensive tuning, including the Logistic Regression equivalent case (α = 0), the Ridge Classifier did not outperform other models like XGBoost (AUC: 0.683 ±0.238). However, we acknowledge that comprehensive hyperparameter tuning should be performed for all models, including XGBoost, to ensure a fair comparison of their optimal performances.
Based on your valid comment on this issue, we added the following paragraph addressing hyperparameter tuning in relation to Figure A1 (page 50):
While our initial comparison using default hyperparameters showed XGBoost performing best (Accuracy: 0.72, AUC: 0.683), we acknowledge a limitation in not performing comprehensive hyperparameter optimization across all models. The relative performance differences seen in Figure A1 might not represent each algorithm's true potential, as models like Random Forest, SVM, and particularly XGBoost are known to be sensitive to hyperparameter settings. Future work should include systematic hyperparameter tuning using techniques such as grid search or Bayesian optimization for all models, especially given our relatively small dataset (n=78) where optimal parameter settings could significantly impact model performance. This would provide a more rigorous comparison and potentially reveal better-performing model configurations than the current default settings.
Round 4
Reviewer 2 Report
Comments and Suggestions for Authors
Thanks for the authors for the continuous responses. I have no more comments now.